# Direct modulation of TRPM8 ion channels by rapamycin and analog macrolide immunosuppressants

**Balázs István Tóth**[1,2]*[†], **Bahar Bazeli**[2,3†], **Annelies Janssens**[2,3], **Erika Lisztes**[1], **Márk Racskó**[1,4], **Balázs Kelemen**[1,2], **Mihály Herczeg**[5], **Tamás Milán Nagy**[6,7], **Katalin E Kövér**[6,8], **Argha Mitra**[9,10], **Attila Borics**[9], **Tamás Bíró**[11], **Thomas Voets**[2,3]*

[1]Laboratory of Cellular and Molecular Physiology, Department of Physiology, Faculty of Medicine, University of Debrecen, Debrecen, Hungary; [2]Laboratory of Ion Channel Research, Department of Cellular and Molecular Medicine, Leuven, Belgium; [3]VIB Center for Brain & Disease Research, Leuven, Belgium; [4]Doctoral School of Molecular Medicine, Faculty of Medicine, University of Debrecen, Debrecen, Hungary; [5]Department of Pharmaceutical Chemistry, University of Debrecen, Debrecen, Hungary; [6]MTA-DE Molecular Recognition and Interaction Research Group, University of Debrecen, Debrecen, Hungary; [7]Department of Chemistry, University of Umeå, Umeå, Sweden; [8]Department of Inorganic and Analytical Chemistry, University of Debrecen, Debrecen, Hungary; [9]Laboratory of Chemical Biology, Institute of Biochemistry, HUN-REN Biological Research Centre, Szeged, Hungary; [10]Theoretical Medicine Doctoral School, Faculty of Medicine, University of Szeged, Szeged, Hungary; [11]Department of Immunology, Faculty of Medicine, University of Debrecen, Debrecen, Hungary

*For correspondence: tibalazs@gmail.com (BIT); thomas.voets@kuleuven.be (TV)

[†]These authors contributed equally to this work

**Competing interest:** The authors declare that no competing interests exist.

## eLife Assessment

This manuscript presents **valuable** findings showing that rapamycin directly activates the cool-sensing ion channel, TRPM8, acting through a different binding site than other small-molecule cooling agents such as menthol. The use of Ca2+-imaging, electrophysiology, and computational biology provides **solid** evidence to support the finding. The authors also present a novel NMR-based method to help identify details of the binding site interactions. In this revised version, some analysis and the presentation have been corrected and improved. Their findings provide insights into TRP channel pharmacology and may indicate previously unknown physiological effects or therapeutic mechanisms of the immunosuppressant, rapamycin.

**Abstract** Rapamycin (sirolimus), a macrolide compound isolated from the bacterium *Streptomyces hygroscopicus*, is widely used as oral medication for the prevention of transplant rejection and the treatment of lymphangioleiomyomatosis. It is also incorporated in coronary stent coatings to prevent restenosis and in topical preparations for the treatment of skin disorders. Rapamycin's in vivo activities are generally ascribed to its binding to the protein FKBP12, leading to potent inhibition of the mechanistic target of rapamycin kinase (mTOR) by the FKBP12-rapamycin complex. The specific rapamycin-induced interaction between domains from mTOR and FKBP12 is also frequently employed in cell biological research, for rapid chemically-induced protein dimerization strategies. Here, we show that rapamycin activates TRPM8, a cation channel expressed in sensory nerve endings that serves as the primary cold sensor in mammals. Using a combination of electrophysiology, Saturation Transfer Triple-Difference (STTD) NMR spectroscopy, and molecular docking-based

targeted mutagenesis, we demonstrate that rapamycin directly binds to human TRPM8. We identify a rapamycin-binding site in the groove between voltage sensor-like domain and the pore domain, distinct from the interaction sites of cooling agents and known TRPM8 agonists menthol and icilin. Related macrolide immunosuppressants act as partial TRPM8 agonists, competing with rapamycin for the same binding site. These findings identify a novel molecular target for rapamycin and provide new insights into the mechanisms of TRPM8 activation, which may assist in the development of therapies targeting this ion channel. Moreover, our findings also indicate that caution is needed when using molecular approaches based on rapamycin-induced dimerization to study ion channel regulation.

## Introduction

TRPM8, a member of the transient receptor potential (TRP) superfamily of cation channels, has been extensively studied in the context of sensory perception and thermoregulation (*Iftinca and Altier, 2020*; *Kashio and Tominaga, 2022*; *Vriens et al., 2014*). It is highly expressed in a subset of somatosensory neurons, where it acts as the principal molecular detector of decreases in temperature (*Bautista et al., 2007*; *Colburn et al., 2007*; *Dhaka et al., 2007*; *McKemy et al., 2002*; *Peier et al., 2002*). TRPM8-deficient mice have pronounced deficits in their response to cool and warm temperatures, do not exhibit cold-induced analgesia, and also fail to detect mild warming of the skin (*Bautista et al., 2007*; *Colburn et al., 2007*; *Dhaka et al., 2007*; *Palkar et al., 2018*). In addition, TRPM8 expression is increased in various malignancies, making it a potential target or biomarker for cancer treatments (*Ochoa et al., 2023*).

Interestingly, TRPM8 is not only activated by cooling, but also by natural or synthetic cooling agents such as menthol or icilin (*McKemy et al., 2002*; *Peier et al., 2002*; *Voets et al., 2004*). Whereas the conformational changes that lead to cold-induced activation of TRPM8 remain largely elusive, the binding sites of menthol analogs and icilin have been resolved in high detail, based on site-directed mutagenesis studies and high-resolution cryo-EM structures (*Bandell et al., 2006*; *Diver et al., 2019*; *Voets et al., 2007*; *Xu et al., 2020*; *Yin et al., 2019*; *Yin et al., 2018*; *Yin et al., 2022*). These studies reveal that menthol binds in the cavity between the four alpha-helices of the voltage sensor-like domain (S1-S4), at a position halfway the membrane, where it interacts with residues in S2 and S4 (*Xu et al., 2020*; *Yin et al., 2018*). Icilin binds in the same cavity, but slightly closer to the cytosolic side of the membrane, in a binding mechanism that is potentiated by cytosolic calcium (*Diver et al., 2019*; *Zhao et al., 2022*). Furthermore, activity of TRPM8 is tightly regulated by the membrane lipid phosphatidylinositol-4,5-bisphosphate ($PIP_2$), which binds to an interfacial cavity formed by the outside surface of S4, S5, and several cytosolic domains (*Diver et al., 2019*; *Rohács et al., 2005*; *Yin et al., 2019*).

Rapamycin is a chemical compound isolated from the bacterium *Streptomyces hygroscopicus*, which was first discovered in a soil sample on Easter Island (Rapa Nui; *Abraham and Wiederrecht, 1996*). Initially described as an antifungal agent (*Vézina et al., 1975*), it was later found to possess potent immunosuppressive properties (*Abraham and Wiederrecht, 1996*). Subsequent research has uncovered its diverse therapeutic potential, ranging from its use in preventing organ transplant rejection and coronary stent restenosis to the treatment of lymphangioleiomyomatosis, skin disorders, cancer, and age-related diseases (*Abraham and Wiederrecht, 1996*; *Ali et al., 2022*; *McCormack et al., 2011*; *Morice et al., 2002*; *Selvarani et al., 2021*; *Swarbrick et al., 2021*). The various biological and therapeutic effects of rapamycin are primarily attributed to inhibition of mechanistic target of rapamycin kinase (mTOR). Rapamycin binds to the protein FKBP12, leading to an inhibitory interaction of the FKBP12-rapamycin complex with the FRB (FKBP12-rapamycin binding) domain of mTOR (*Abraham and Wiederrecht, 1996*). The ability of rapamycin and rapamycin analogs to rapidly and efficiently bring FKBP12 and the FRB domain in close contact has also been successfully engineered for the rapid chemical dimerization and translocation of signaling enzymes or receptors (*Muthuswamy et al., 1999*), including (TRP) ion channels (*Suh et al., 2006*; *Varnai et al., 2006*).

While performing experiments involving rapamycin-induced membrane translocation of a phospholipid phosphatase, we serendipitously discovered that rapamycin by itself potently activates TRPM8, both in TRPM8-expressing HEK cells and in mouse somatosensory neurons. Using whole-cell and inside-out patch-clamp recordings and Saturation Transfer Triple-Difference (STTD) NMR

spectroscopy, we demonstrate that channel activation is caused by direct binding of rapamycin to TRPM8, rather than via an effect on mTOR activity. Based on site-directed mutagenesis guided by molecular docking, we provide evidence for a rapamycin-interaction site in the groove between S4 and S5, distinct from the known interaction sites for menthol or icilin. We also show that related macrolides such as the immunosuppressant everolimus act as partial TRPM8 agonists and compete with rapamycin for the same binding site. Our results identify TRPM8 as a novel molecular target for rapamycin and provide new insights into the mechanisms of TRPM8 activation, which may assist in the development of drugs targeting this ion channel. Finally, our results imply that direct activation of TRPM8 may need to be taken into account when using rapamycin-based dimerization approaches to study cellular processes in vitro and in vivo.

## Results

### Rapamycin activates TRPM8

When performing Fura-2-based intracellular $Ca^{2+}$ concentration ($[Ca^{2+}]_i$) measurements in HEK293 cells stably expressing human TRPM8 channels (HEK-M8 cells), we observed that rapamycin (10 µM) caused a robust increase in $[Ca^{2+}]_i$, comparable in amplitude to the response evoked by the prototypical TRPM8 agonist menthol at a concentration of 50 µM (**Figure 1A**). Responses to both rapamycin and menthol were fully inhibited by the specific TRPM8 antagonist AMTB N-(3-aminopropyl)–2-[(3-methylphenyl) methoxy] -N-(2-thienylmethyl) benzamide hydrochloride; 2 µM (**Lashinger et al., 2008**; **Figure 1A**). The effect of rapamycin was concentration-dependent, with saturating responses at concentrations ≥10 µM and an $EC_{50}$ value of 3.8±2.0 µM (**Figure 1B**). Similar results were obtained using a 96-well plate-based assay, where we obtained an $EC_{50}$ value of 6.0±0.3 µM at room temperature, which shifted to 10.1±0.2 µM at 37 °C (**Figure 1—figure supplement 1**). Likewise, in whole-cell patch-clamp measurements on HEK-M8 cells at room temperature, rapamycin (10 µM) evoked robust TRPM8 currents, which rapidly returned to baseline upon washout of rapamycin and were fully inhibited by AMTB (2 µM; **Figure 1C**). Half-maximal activation of whole-cell currents at + 120 mV was obtained at a rapamycin concentration of 4.5±1.8 µM (**Figure 1D**). Importantly, menthol and rapamycin did not evoke any detectable calcium signal or current increase in non-transfected HEK293 cells (**Figure 1—figure supplement 2** and Figure 3C).

Next, we tested other TRP channels involved in chemo- and thermosensation (**Bamps et al., 2021**) for their sensitivity to rapamycin (**Figure 1—figure supplement 3**). We did not observe any sizeable current responses to 30 µM rapamycin in HEK293 cells expressing TRPA1, TRPV1, or TRPM3, whereas large currents were measured in response to their respective agonists allyl isothiocyanate (AITC; 373±173 pA/pF at +120 mV; N=5), capsaicin (1733±677 pA/pF at +120 mV; N=4) and pregnenolone sulfate (PS; 270±71 pA/pF at +120 mV; N=5), respectively.

TRPM8 is expressed in a small subpopulation (5–10%) of somatosensory neurons, where it plays a central role in detection of cool and warm temperature (**Bautista et al., 2007**; **Colburn et al., 2007**; **Dhaka et al., 2007**; **McKemy et al., 2002**; **Palkar et al., 2018**; **Peier et al., 2002**). To evaluate whether rapamycin activates TRPM8 in these neurons, we performed Fura-2-based calcium imaging experiments on somatosensory neurons isolated from the dorsal root and trigeminal ganglia (DRG and TG) of wild type ($Trpm8^{+/+}$) and $Trpm8^{-/-}$ mice (**Figure 1E**). In wild-type neurons, approximately 10% of DRG and TG neurons showed a robust calcium response to rapamycin (10 µM). The large majority (95%) of these rapamycin-responsive neurons also responded to menthol (50 µM). Importantly, responses to rapamycin were largely eliminated in DRG and TG neurons from $Trpm8^{-/-}$ animals, indicating that TRPM8 mediates rapamycin responses in sensory neurons (**Figure 1E and F**).

Notably, a subset (~7%) of both $Trpm8^{+/+}$ and $Trpm8^{-/-}$ neurons responded to menthol but not to rapamycin (**Figure 1F**). These findings are in line with earlier studies demonstrating that menthol acts as an agonist of not only TRPM8 but also TRPA1 (**Karashima et al., 2007**; **Xiao et al., 2008**). In agreement herewith, we found that 92 ± 7% of the rapamycin-insensitive menthol-responsive neurons were also activated by the TRPA1 agonist cinnamaldehyde. Moreover, in experiments using the TRPM8 antagonist AMTB, we found that 2 µM AMTB fully inhibited the calcium responses to menthol and rapamycin in rapamycin-responsive $Trpm8^{+/+}$ neurons, whereas menthol responses in rapamycin-insensitive neurons were not inhibited by AMTB (**Figure 1—figure supplement 4**). Taken together,

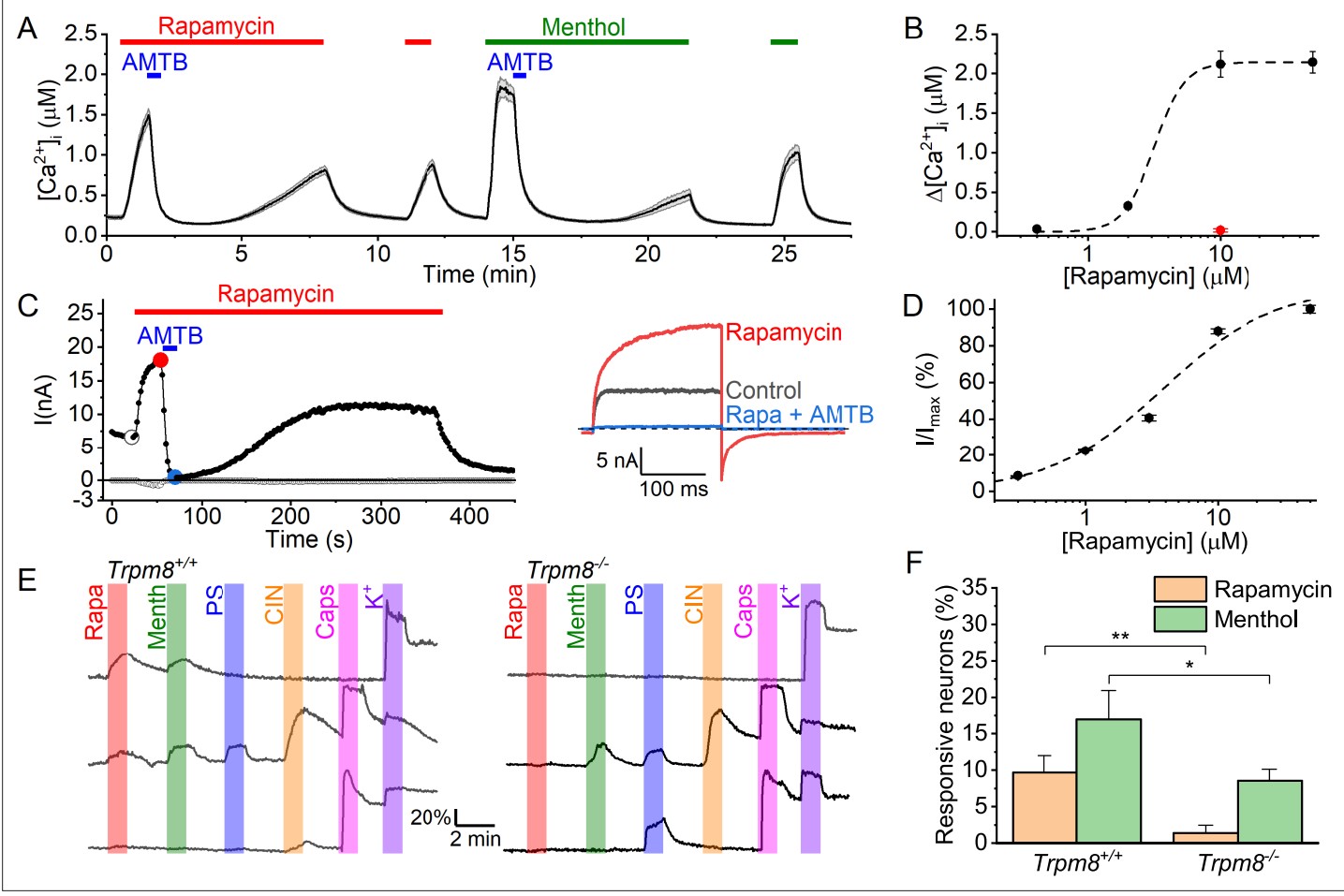

**Figure 1.** Rapamycin activates TRPM8 in HEK293 cells and sensory neurons. (**A**) Time course of the intracellular calcium concentration in HEK293 cells expressing TRPM8, showing robust responses to rapamycin (10 μM) and menthol (50 μM), and inhibition of the responses by AMTB (2 μM). Shown are mean ± SEM, N=63 cells. (**B**) Concentration dependence of rapamycin-evoked calcium responses. The dashed line represents the best fit using a Hill equation (EC$_{50}$=3.1; n$_H$ = 3.8). Shown are mean ± SEM, N=52–95 cells/concentration. The red symbol indicates the lack of response to rapamycin (10 μM) in non-transfected HEK293 cells (N=23). (**C**) *Left,* time course of whole-cell currents in HEK293 cells expressing TRPM8 evoked by repetitive voltage steps to +120 and –80 mV, showing the activation of outwardly rectifying currents by rapamycin (10 μM) and inhibition by AMTB (2 μM). *Right,* voltage steps recorded at the indicated time points. (**D**) Concentration dependence of rapamycin-evoked whole-cell currents at +120 mV. The dashed line represents the best fit using a Hill equation (EC$_{50}$=3.8; n$_H$ = 1.0). Shown are mean ± SEM, N=5 cells per concentration. (**E**) Examples of calcium signals in individual DRG neurons from Trpm8$^{+/+}$ and Trpm8$^{-/-}$ mice in response to rapamycin (Rapa; 10 μM), menthol (Menth; 50 μM), pregnenolone sulphate (PS; 40 μM), cinnamaldehyde (CIN; 10 μM), capsaicin (Caps; 100 nM), or high K$^+$ (50 Mm). (**F**) Fraction of sensory neurons from Trpm8$^{+/+}$ and Trpm8$^{-/-}$ mice that responded to rapamycin and menthol. *p<0.05 and **p<0.01.

The online version of this article includes the following source data and figure supplement(s) for figure 1:

**Source data 1.** Raw values used for plots in *Figure 1*.

**Figure supplement 1.** Rapamycin activates hTRPM8 in a concentration-dependent manner.

**Figure supplement 1—source data 1.** Raw values used for plots in *Figure 1—figure supplement 1*.

**Figure supplement 2.** Rapamycin does not evoke calcium signals in naïve HEK293 cells.

**Figure supplement 2—source data 1.** Raw values used for plots in *Figure 1—figure supplement 2*.

**Figure supplement 3.** Rapamycin does not activate TRPA1, TRPM3, or TRPV1.

**Figure supplement 4.** Rapamycin allows distinguishing between TRPM8-mediated and TRPM8-independent menthol responding sensory neurons.

these findings indicate that rapamycin selectively activates TRPM8-positive sensory neurons, whereas menthol is less selective, exciting both TRPM8-positive and TRPA1-positive neurons.

Taken together, these data show that rapamycin activates TRPM8 in a heterologous expression system and in sensory neurons and indicate that rapamycin is a more selective tool than menthol to identify TRPM8-positive sensory neurons.

## Rapamycin is an agonist ligand directly binding to TRPM8

We initially considered the possibility that the rapamycin-induced activation of TRPM8 in both HEK293 cells and sensory neurons could occur downstream of its effects on mTOR. However, the rapid onset and reversibility of rapamycin's effect on TRPM8 currents, and the observation that low micromolar concentrations of rapamycin were required to induce detectable current activation (compared to the nanomolar concentrations of rapamycin required for mTOR inhibition; *Abraham and Wiederrecht, 1996*; *Varnai et al., 2006*) spoke against an mTOR-dependent effect. Notably, rapamycin also activated TRPM8 when applied to the cytoplasmic surface of excised inside-out patches (*Figure 2A*), indicating that it acts on TRPM8 in a membrane-delimited manner. Based on these observations, we examined whether rapamycin could exert an agonistic action on TRPM8 via direct binding of the compound to the channel protein.

To test a direct molecular interaction between rapamycin and TRPM8, and to obtain insights into the parts of the rapamycin molecule involved in the interaction, we developed the Saturation Transfer Triple-Difference NMR (STTD-NMR) method. (*Figure 2B–E*). This new approach allowed us to obtain a clean $^1$H-STTD NMR spectrum, eliminating interference from any non-specific interactions present in complex cellular systems. The basic principle of the classical, one-dimensional saturation transfer difference $^1$H NMR (1D STD-NMR) is that resonances are saturated by selective irradiation, and this saturation spreads over the whole protein molecule and its bound ligands via spin diffusion (*Meyer and Peters, 2003*). The signal attenuation of ligand $^1$H resonances upon binding is evident in the saturation transfer difference (STD) spectrum, which is obtained by subtraction of two spectra, one recorded with and the other without the saturation of protein resonances. We calculated the difference of STD spectra (double-difference STD, STDD; *Claasen et al., 2005*), one recorded on a cellular sample with rapamycin added (sample 1, STD-1 in *Figure 2B and C*) and the other without rapamycin (sample 2, STD-2 in *Figure 2B and C*). The resulting double-difference spectrum (STDD-1 in *Figure 2C*) reports both TRPM8-specific and non-specific binding interactions of rapamycin. To remove the STD signals arising from non-specific interactions of rapamycin, we recorded two more STD spectra on cells lacking TRPM8 expression (naïve HEK293 cells), namely in the presence (sample 3, STD-3) and absence (sample 4, STD-4) of rapamycin, yielding the second double-difference (STDD-2) spectrum (*Figure 2C*). Finally, the difference of STDD-1 and STDD-2 spectra yielded the triple-difference spectrum (STTD), which is free of interfering signals arising from any non-specific interactions (*Figure 2C*). To address the effect of the inherent variability of cellular samples on peak heights, STD effects were normalized based on the comparison of independent $^1$H experiments (*Figure 2—figure supplement 1*). Three STTD replicates were computed, unambiguously confirming direct binding to TRPM8 in two datasets (*Figure 2—figure supplement 2A and B*). The signals assigned in the STTD spectra (numbered according to the schematic structure of rapamycin; *Figure 2D*) unveil the rapamycin H atoms that correspond to the binding epitopes of the molecule to TRPM8 (*Figure 2E*). Taken together, these findings indicate that rapamycin binds to TRPM8, acting as a direct channel agonist independently of mTOR.

In earlier work, we have classified TRPM8 agonists into two types based on their effect on the channel's gating kinetics (*Janssens et al., 2016*). According to this classification, Type I agonists such as menthol induce a slowing of the gating kinetics, which is most prominently observed as slowly deactivating tail currents following repolarization, whereas Type II agonists such as AITC cause an acceleration of the kinetics of channel activation upon depolarization, with little or no effect on the kinetics of deactivating tail currents. These differences in kinetics can be explained by a gating model where type I agonists cause a relative stabilization of the open state, whereas type II agonists rather destabilize the closed state (*Janssens et al., 2016*). Whole-cell current recordings in HEK-M8 cells revealed a pronounced effect of rapamycin on the kinetics of current relaxation in response to voltage steps (see e.g. *Figure 1C*; *Figure 2—figure supplement 3*). In particular, rapamycin caused a concentration-dependent slowing of the time course of voltage-dependent activation and deactivation, resulting in

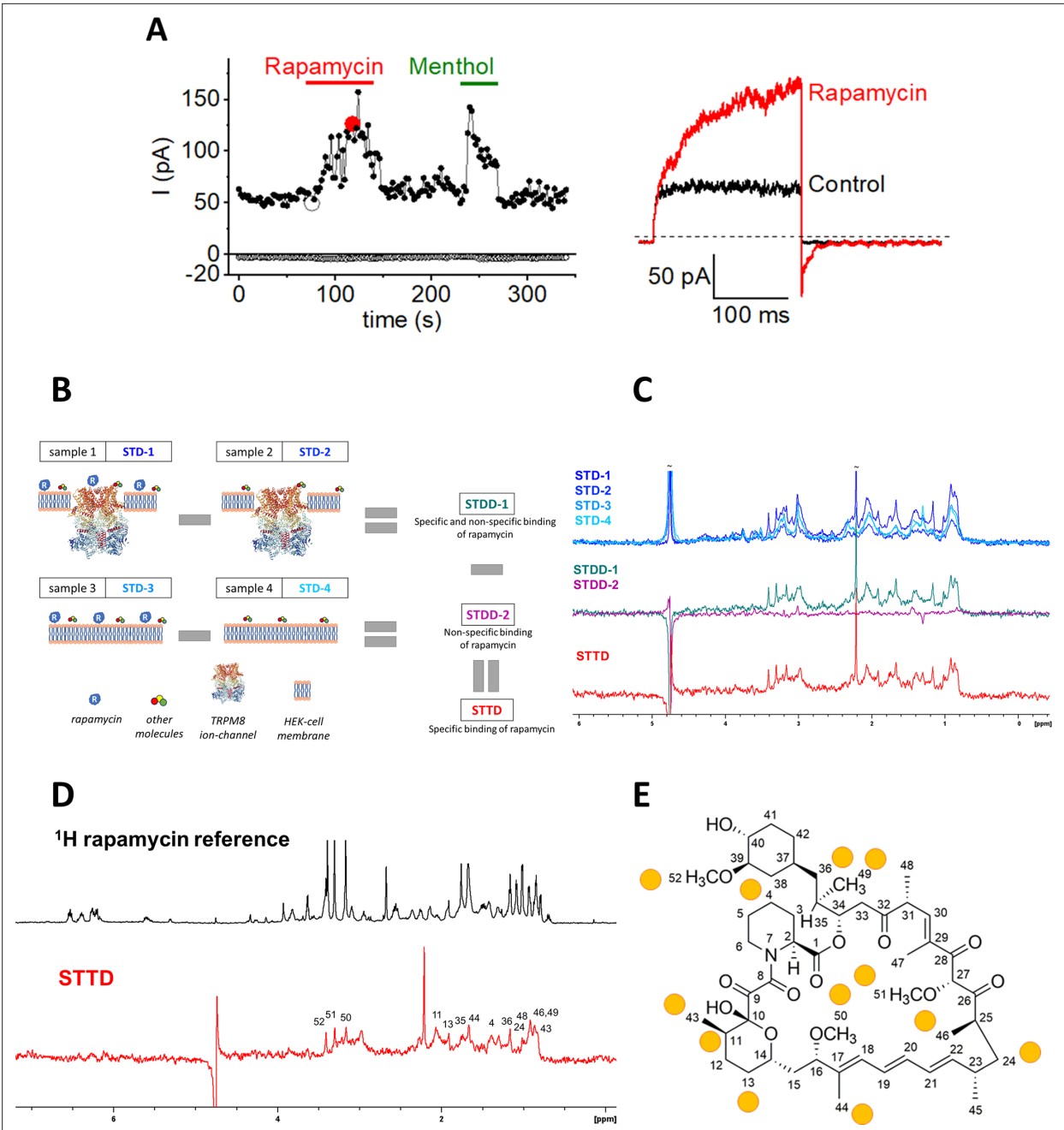

**Figure 2.** Direct interaction between rapamycin and TRPM8. (**A**) *Left,* time course of currents in a cell-free inside-out patch pulled from a HEK293 cell expressing TRPM8 evoked by repetitive voltage steps to +80 and –80 mV, showing the activation of outwardly rectifying currents by rapamycin (10 μM) and menthol (50 μM) applied from the cytosolic side. *Right,* voltage steps recorded at the indicated time points. This example is representative of five similar experiments. (**B**) Cartoon representing the steps to obtain the direct interaction of rapamycin with TRPM8. Individual STD spectra were recorded on different sample compositions, then non-specific interactions were filtered out with multiple subtractions resulting in the double difference spectra (STDD-1,2) and the final triple difference spectrum (STTD) shown on the right side. (**C**) The four STD spectra (STD-1,2,3,4) are overlaid for comparison. Double difference spectra were computed from the respective STD pairs showing the specific and non-specific binding of rapamycin (STDD-1, green) and the non-specific binding of rapamycin alone (STDD-2, purple). All non-specific interactions were filtered out in the final STTD spectrum (red) computed by subtracting STDD-2 from STDD-1. Here, only one experimental set is shown (dataset B), spectra on the replicates can be found in the supplementary (*Figure 2—figure supplements 1 and 2*). (**D**) Rapamycin resonances involved in the direct interaction with TRPM8 were assigned by the comparison of the reference $^1$H NMR spectra of rapamycin (black) with the computed STDD effects (red). The reference spectrum was recorded on a 0.2 mM rapamycin sample in a D$_2$O buffer solution at 25 °C. The number of scans was 256 and a watergate sequence was used to suppress the residual water signal. (**E**) Hydrogen atoms involved in the interaction with TRPM8 are mapped to the structure of rapamycin (yellow circles).

*Figure 2 continued on next page*

*Figure 2 continued*

The online version of this article includes the following source data and figure supplement(s) for figure 2:

**Source data 1.** Raw values used for plots in *Figure 2*.

**Figure supplement 1.** The variability of peak heights in $^1$H NMR experiments and the normalization procedure.

**Figure supplement 2.** Replicates of STTD measurements.

**Figure supplement 3.** Rapamycin acts as a type I agonist.

**Figure supplement 3—source data 1.** Raw values used for plots in *Figure 2—figure supplement 3*.

**Figure supplement 4.** Additive effects of Rapamycin and menthol and TRPM8 deactivation.

**Figure supplement 4—source data 1.** Raw values used for plots in *Figure 2—figure supplement 4*.

long-lasting tail currents upon repolarization from a strongly depolarizing voltage step to +120 mV, and these effects were more pronounced than for menthol (*Figure 2—figure supplement 3*). Overall, these characteristics classify rapamycin as a type I agonist, stabilizing the channel's open conformations. Notably, rapamycin and menthol had a synergistic effect on channel gating: application of menthol (50 µM) in the continued presence of rapamycin (10 µM) caused a further slowing down of channel deactivation (*Figure 2—figure supplement 4*).

## Rapamycin binds to a unique binding site on TRPM8

Site-directed mutagenesis studies and high-resolution cryo-EM structures have delineated binding sites for several TRPM8 ligands, including agonists such as icilin, AITC, cryosim-3 and the menthol analog WS-12, as well as antagonists such as AMTB or TC-I 2014 (*Bandell et al., 2006*; *Diver et al., 2019*; *Voets et al., 2007*; *Yin et al., 2019*; *Yin et al., 2018*; *Yin et al., 2022*). These compounds all bind in the cavity between the four alpha-helices of the voltage sensor-like domain (S1-S4), with the exception of the type II agonist AITC, which binds outside the voltage sensor-like domain at a site between S3 and the S4-S5 linker (*Yin et al., 2022*).

Since rapamycin, like menthol, acts as a type I agonist, and both compounds contain a substituted cyclohexyl moiety, we initially examined the possibility that rapamycin would interact with the menthol-binding site. However, considering the much larger size of rapamycin (molecular weight of 914.2 g/mol) compared to agonists such as menthol, WS-12, or cryosim-3 (molecular weights of 154.3, 260.4, and 289.4 g/mol, respectively), binding of rapamycin in the relatively narrow cavity in the voltage sensor-like domain appeared unlikely. In agreement with this notion, initial experiments revealed that mutations in the menthol binding site that strongly reduce responses to menthol, icilin, Cooling Agent 10, WS-3, or cryosim-3 (Y745H and R842H; *Bandell et al., 2006*; *Voets et al., 2007*; *Malkia et al., 2009*; *Beccari et al., 2017*; *Plaza-Cayón et al., 2022*; *Yin et al., 2022*; *Palchevskyi et al., 2023*) did not affect the channel's sensitivity to rapamycin (see *Figure 3*). Therefore, we hypothesized that rapamycin interacts at a different ligand binding site on TRPM8.

To identify potential rapamycin interaction sites, we performed blind, multistep dockings of rapamycin to a model of the human TRPM8 channel based on a cryo-EM structure of mouse TRPM8. Initial pilot dockings revealed several potential rapamycin interaction sites, either at the bottom of the voltage sensor-like domain or in the groove between the voltage sensor-like domain and the pore region (*Figure 3—figure supplement 1*). Based on these initial results, we made a series of point mutations at strategic residues in these potential rapamycin interaction sites and used Fura-2-based $[Ca^{2+}]_i$ measurements to test these mutants for their responses to menthol (50 µM) and rapamycin (10 µM; *Figure 3A–C*). All mutant channels yielded robust calcium responses to at least one of the two agonists, indicating that they expressed as functional channels (*Figure 3C*). Wild type and most mutants showed maximal responses ($\Delta[Ca^{2+}]_i$) to at least one of the agonists of >1 µM. One exception was mutant F847A, which did not respond to menthol but showed a consistent but relatively small ($\Delta[Ca^{2+}]_i \approx 200$ nM) to rapamycin (*Figure 3C*, *Figure 3—figure supplement 2*).

To assess potential changes in sensitivity to rapamycin or menthol, we calculated the rapamycin response ratio, defined as the calcium response to rapamycin over the sum of the calcium responses to rapamycin and menthol ($\Delta[Ca^{2+}]_{i,rapamycin}/(\Delta[Ca^{2+}]_{i,rapamycin} + \Delta[Ca^{2+}]_{i,menthol})$). As such, mutants that respond to rapamycin but fully lack a response to menthol have a response ratio of 1, while mutants that respond to menthol but fail to respond to rapamycin have a response ratio of 0. For the wild-type

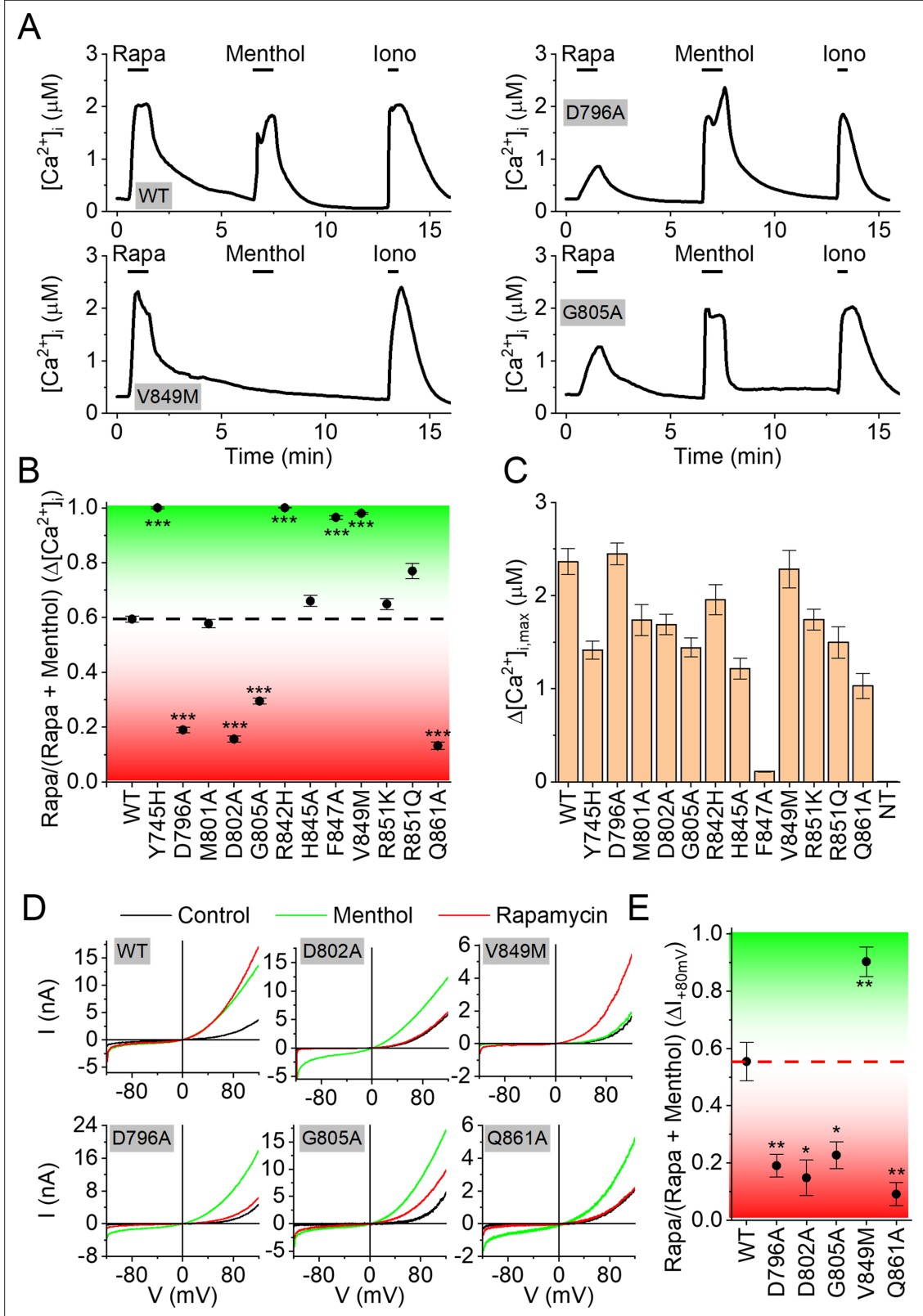

**Figure 3.** TRPM8 residues involved in the interaction with rapamycin and menthol. (**A**) Representative time courses of the intracellular calcium concentration in HEK293 cells expressing wild type TRPM8 or the indicated mutants, when stimulated with rapamycin (10 μM), menthol (50 μM), and the calcium ionophore ionomycin (2 μM). (**B**) Quantification of the relative calcium response to rapamycin and menthol for wild type and the indicated TRPM8 mutants. Values indicate the ratio between the calcium response amplitude to rapamycin, divided by the sum of the responses to rapamycin and

*Figure 3 continued on next page*

*Figure 3 continued*

menthol. The dotted line represents the mean value for wild type TRPM8. Values above this line (in yellow) indicate a relative reduction in the response to menthol, whereas values below the line (cyan) indicate a relative reduction in the response to rapamycin. (**C**) Amplitude of the calcium response to the agonist (menthol or rapamycin) that gave the largest response for wild type and the indicated TRPM8 mutants. Data in B and C represent mean ± SEM, N=34–156/group. (**D**) Whole-cell current-voltage relations for the currents in control, and in the presence of rapamycin (10 μM) or menthol (50 μM) in HEK293 cells expressing wild type TRPM8 or the indicated mutants. (**E**) *, **, *** indicate p<0.05, p<0.01, and p<0.001 compared to WT. (**F**) Quantification of the relative current response to rapamycin and menthol for wild type and the indicated TRPM8 mutants. Values indicate the ratio between the current amplitude increase at +80 mV to rapamycin, divided by the sum of the responses to rapamycin and menthol. The dotted line represents the mean value for wild type TRPM8. Values above this line (in yellow) indicate a relative reduction in the response to menthol, whereas values below the line (cyan) indicate a relative reduction in the response to rapamycin. Data in represent mean ± SEM; N=5–8 per mutant.

The online version of this article includes the following source data and figure supplement(s) for figure 3:

**Source data 1.** Raw values used for plots in *Figure 3*.

**Figure supplement 1.** Potential binding sites and poses of rapamycin on TRPM8 obtained from blind pilot dockings.

**Figure supplement 2.** Rapamycin, but not menthol, moderately activates HEK cells expressing TRPM8[F847A].

**Figure supplement 2—source data 1.** Raw values used for plots in *Figure 3—figure supplement 2*.

channel, which robustly responds to both agonists, this analysis yielded a response ratio of 0.59±0.02 (*Figure 3B*).

Several mutants yielded a significantly higher response ratio compared to wild type, indicating reduced menthol sensitivity. These included the previously described mutants Y745H and R842H (*Bandell et al., 2006*; *Voets et al., 2007*), which show robust responses to rapamycin but no responses to menthol (response ratios >0.95; *Figure 3A and B*). These residues, located in the middle of S1 and S4, respectively, point towards the center of the cavity within the voltage sensor-like domain, where they interact with ligands such as cryosim-3 and the menthol analog WS-12. Mutants R851Q and V849M, located at the cytosolic entrance to the cavity within the voltage sensor-like domain, also show a significantly reduced menthol response combined with a robust rapamycin response (response ratios of 0.77±0.03, and 0.98±0.02, respectively; *Figure 3A and B*). The results from these mutations are fully in line with the notion that menthol activates TRPM8 by binding in the middle of the cavity within the voltage sensor-like domain, and that mutations in the binding site or at the entrance of the cavity affect menthol activation. The observation that rapamycin activation is not noticeably affected in all these mutants confirms that the rapamycin and menthol binding sites are separate and also argues against a potential rapamycin binding site at the lower part of the voltage sensor-like domain.

Oppositely, we identified four mutants with a significantly lower rapamycin response ratio compared to wild type (response ratios <0.3), indicating reduced rapamycin sensitivity. These include D796A (located in the S2-S3 linker), D802A and G805A (located in S3), and Q861A (located in the S5 pore helix; *Figure 3A and B*). The strongly reduced sensitivity of these mutants to rapamycin was confirmed in whole-cell patch-clamp recordings, revealing robust current responses to menthol but very limited responses to rapamycin (*Figure 3D and E*). These results further confirm that rapamycin binds directly to the TRPM8 channel and are in agreement with a model where rapamycin binds in the groove between the voltage sensor-like domain and the pore region. These results and the publication of the cryo-EM structure of the mouse TRPM8 during our experiments prompted the further refinement of dockings, resulting in the rapamycin binding site illustrated in *Figure 4*, *Figure 4—figure supplement 1* (the pdb-file of the model is available via https://doi.org/10.5281/zenodo.17143202). In this model, rapamycin interacts with residues from a single subunit. Note that this proposed mode of interaction also largely aligns with the results from the 1H-STTD NMR study, with several of the H atoms that show saturation transfer being effectively in close contact with the channel protein (*Figure 4—figure supplement 2*).

Mutants D802A and G805A mutations were originally identified due to their ability to reduce icilin-induced activation of TRPM8 (*Chuang et al., 2004*). Interestingly, structural studies indicated that these residues are not directly involved in binding icilin. Instead, D802 is involved in calcium binding, which strongly potentiates activation of TRPM8 by icilin, while the flexibility invoked by a glycine at position 805 allows enlargement of the VSLD cavity to accommodate the hydroxyphenyl moiety of icilin (*Yin et al., 2019*). We therefore tested whether rapamycin activation is also modulated by intracellular calcium, by using UV flash-induced release of caged $Ca^{2+}$ during whole-cell current recordings (*Mahieu et al., 2010*). In these experiments, a 1 ms UV flash was applied during a 800 ms voltage step

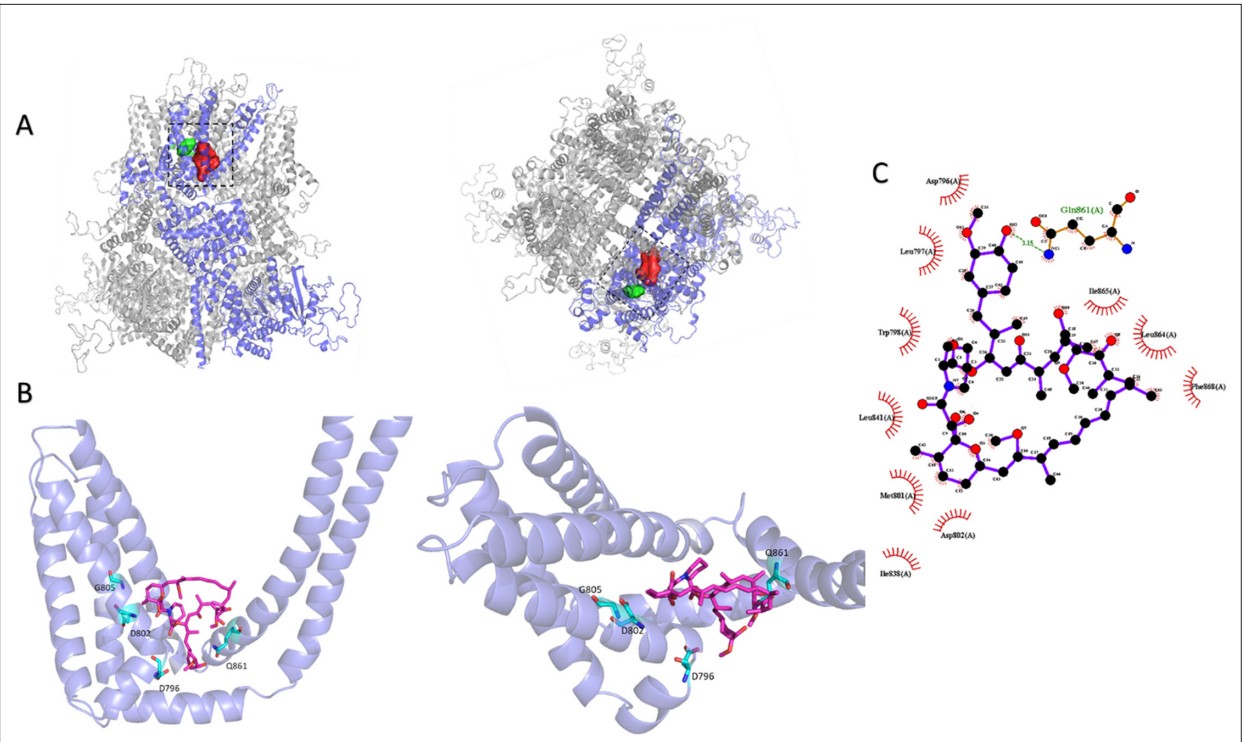

**Figure 4.** Structural model showing the distinct interaction sites for menthol and rapamycin. (**A**) Side view (*left*) and top view (*right*) of TRPM8, showing the known interaction site for menthol (green) and the proposed rapamycin interaction site (red) based on our present molecular docking and mutagenesis studies. (**B**) Closer view of rapamycin docked onto the TRPM8 structure. Amino acid residues that, when mutated, influence rapamycin responses are indicated in green. (**C**) 2D projection of interactions between rapamycin and TRPM8 created using Ligplot⁺.

The online version of this article includes the following figure supplement(s) for figure 4:

**Figure supplement 1.** Structural model indicating side chains for all mutations, colored by menthol (green)/rapamycin (red) selectivity.

**Figure supplement 2.** Visualizing and comparing the STTD NMR results with the predicted rapamycin binding pocket.

to +80 mV, either in the absence of ligand or in the presence of rapamycin (10 µM) or icilin (10 µM). The UV flash caused a rapid increase in intracellular $Ca^{2+}$, as indicated by the change in Fura-FF fluorescence ratio (*Figure 5A*). In line with our earlier work (*Mahieu et al., 2010*), at low cytosolic calcium, icilin evokes a sizable current response, which is robustly potentiated of inward and outward currents in the presence of icilin (*Figure 5A–D*). In contrast, currents recorded in the absence of ligand or in the presence of rapamycin were not directly affected by flash-induced calcium uncaging (*Figure 5A–D*). These results demonstrate that rapamycin activation of TRPM8 is not calcium-dependent and indicate that the reduced rapamycin sensitivity of the D802A mutant is unrelated to this residue's involvement in $Ca^{2+}$ binding.

## Effect of rapamycin analogs on TRPM8

According to the proposed binding mode (*Figure 4*), the substituted cyclohexane ring (*trans*-2-methoxycyclohexan-1-ol) of rapamycin is tightly involved in the interaction with TRPM8. In particular, the model predicts a hydrogen bond formed between Q861 and the hydroxyl group on the cyclohexane ring (i.e. on carbon 40 in the rapamycin structure). A number of rapamycin analogs, also known as rapalogs *Lamming et al., 2013*, have been developed as immunosuppressants, in which the hydroxyl group on the cyclohexane ring has been substituted by other functional groups. When tested using Fura-2-based calcium imaging, we found that everolimus, zotarolimus, ridaforolimus, and temsirolimus were much less effective than rapamycin in activating TRPM8, with amplitudes that were <10% of that to rapamycin at a concentration of 10 µM (*Figure 6A and B*). The efficacy order was rapamycin >>zotarolimus > ridaforolimus >everolimus > temsirolimus. These results reveal the

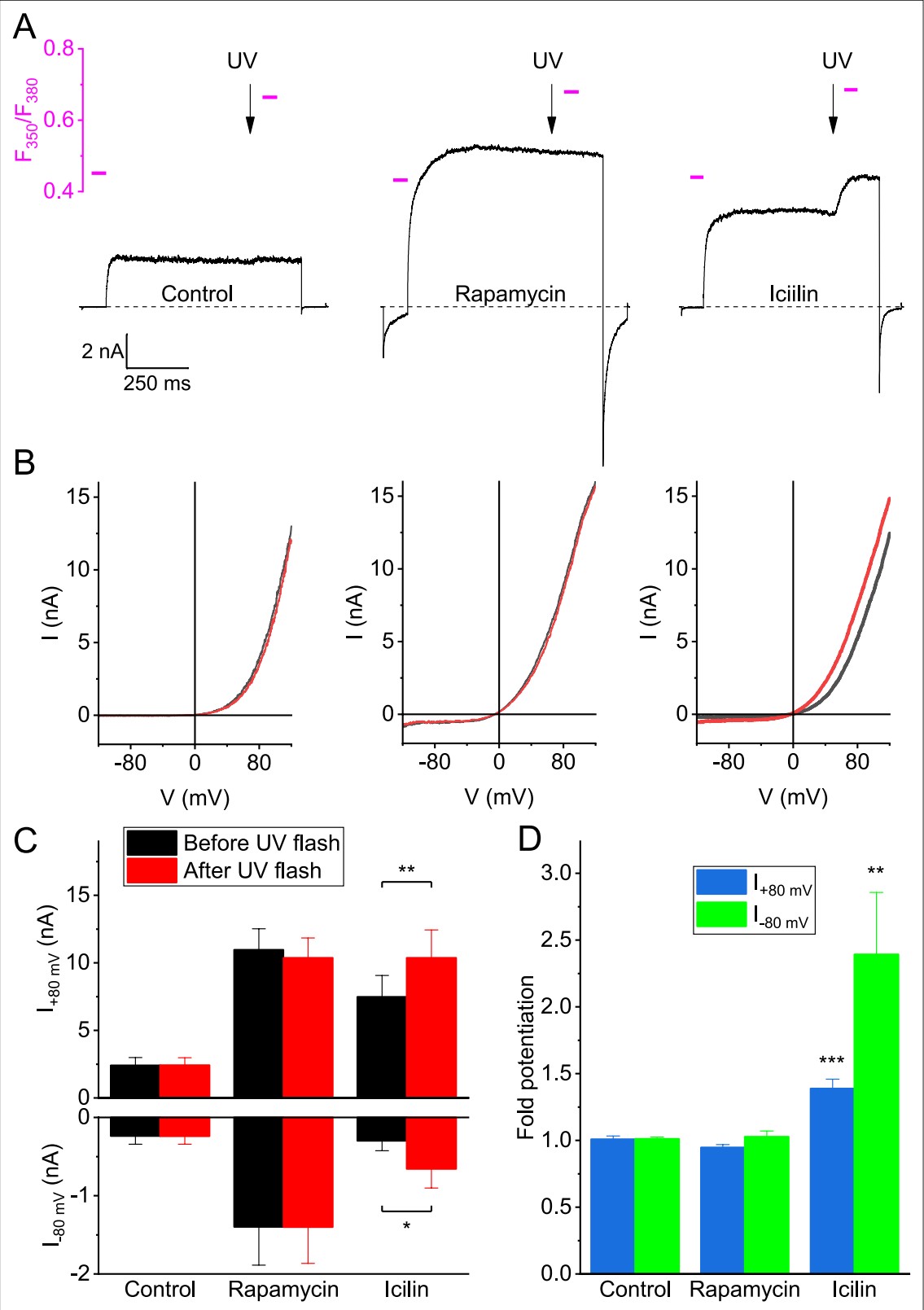

**Figure 5.** Activation of TRPM8 by rapamycin is independent of intracellular calcium. (**A**) Whole-cell currents during 800 ms voltage steps from −80 to +80 mV under control conditions and in the presence of rapamycin (10 μM) or icilin (10 μM). At the time points indicated by the arrows, a 1 ms UV flash was applied, leading to a rapid increase in intracellular calcium. Magenta lines and scale bar indicate Fura-FF fluorescence ratios at the indicated time points. (**B**) Whole-cell current-voltage relations measured during voltage ramps 2 s before and 1 s after the UV flashes shown in panel A.

*Figure 5 continued on next page*

*Figure 5 continued*

(**C**) Current amplitudes at +80 and –80 mV before and after UV uncaging of calcium, under control conditions and in the presence of rapamycin or icilin. (**D**) Quantification of the relative potentiation of inward and outward currents following UV uncaging of calcium. Data in C and D represent the mean ± SEM from five experiments. *,**, ***: p<0.05 in a paired t-test comparing currents before and after UV flash.

The online version of this article includes the following source data for figure 5:

**Source data 1.** Raw values used for plots in *Figure 5*.

importance of the substituted cyclohexane ring, and in particular the hydroxyl group on carbon 40, for the activation of TRPM8 by rapamycin.

The calcineurin inhibitors FK506 (tacrolimus) and its structural analog ascomycin contain the same substituted cyclohexane ring (*trans*-2-methoxycyclohexan-1-ol) as rapamycin, but have a different, smaller macrolactam ring than rapamycin and the rapalogs (*Figure 6A*). Confirming earlier research (*Arcas et al., 2019*), we found that FK506 evoked robust calcium responses in HEK-M8 cells, albeit less potently than rapamycin (*Figure 6B*). We also measured substantial responses to ascomycin, which were intermediate between the responses to FK506 and to the most effective rapalog zotarolimus (*Figure 6B*). Pimecrolimus, a derivative of ascomycin in which a chloride substitutes for the corresponding hydroxyl group on the cyclohexane ring, did not evoke any detectable calcium response

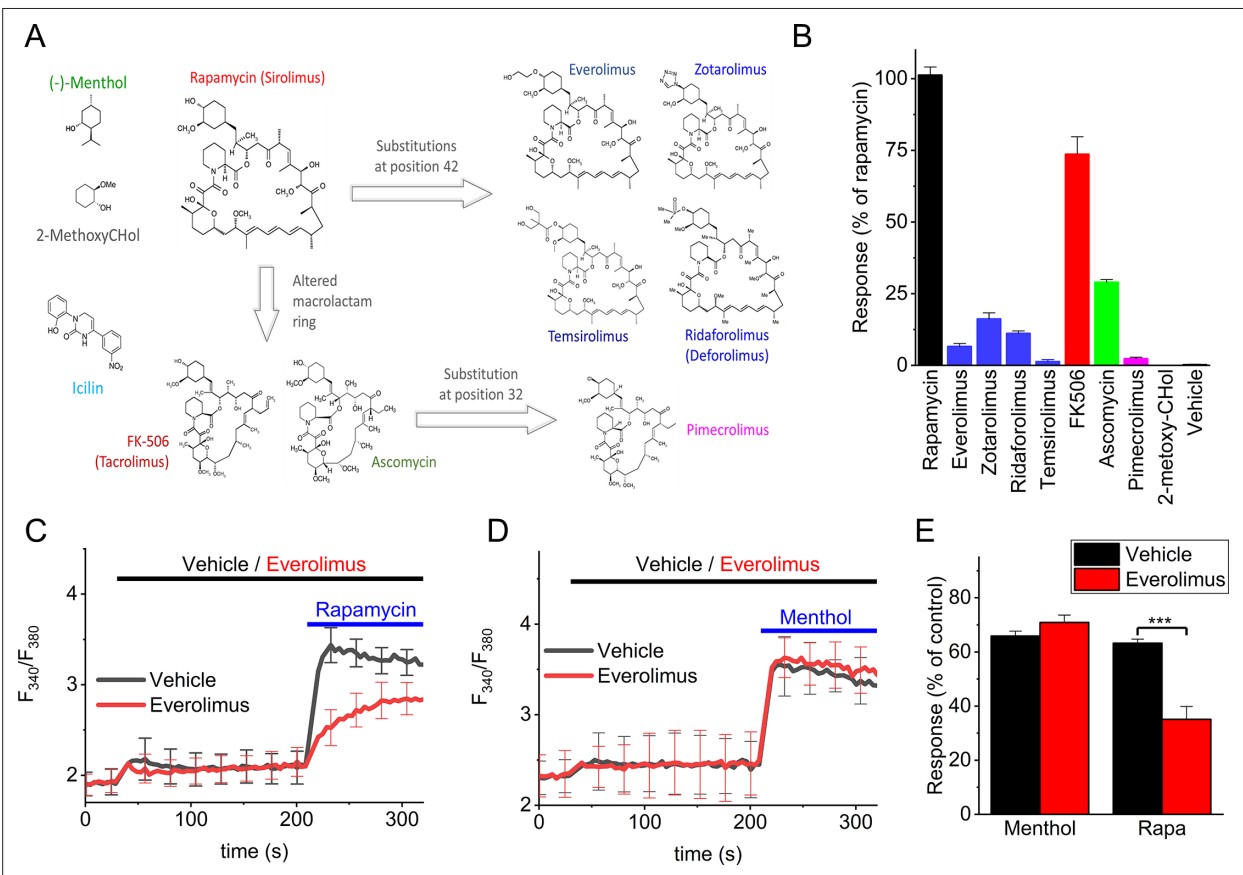

**Figure 6.** Effect of rapamycin and macrolide analogs on TRPM8. (**A**) Overview of the macrolides tested in this study. (**B**) Relative calcium response to rapamycin and the indicated analogs tested at 10 µM. N≥6 in each group. (**C**) Fura2-based calcium response to rapamycin (10 µM) in the presence of everolimus (10 µM) or vehicle. N=6 in each group. (**D**) Fura2-based calcium response to menthol (50 µM) in the presence of everolimus (10 µM) or vehicle. N=6 in each group. (**E**) Summary of the responses to menthol (50 µM) and rapamycin (10 µM) in the absence or presence of everolimus (10 µM). Responses were normalized to the response to a saturating concentration of menthol (300 µM). N=6 in each group. ***: p<0.001.

The online version of this article includes the following source data and figure supplement(s) for figure 6:

**Source data 1.** Raw values used for plots in *Figure 6*.

**Figure supplement 1.** Molecular docking of rapamycin and everolimus to the groove between voltage sensor-like domain and the pore domain of TRPM8.

(*Figure 6A and B*). Taken together, these findings demonstrate that rapamycin is the most effective of the tested macrolides and illustrate the importance of the hydroxyl group at position 40 on the cyclohexane ring of rapamycin for activating TRPM8.

To investigate whether the substituted cyclohexane ring of rapamycin by itself is sufficient to evoke TRPM8 activation, we synthesized and tested *trans*-2-methoxycyclohexan-1-ol (2-methoxy-CHol). However, we could not detect any TRPM8 responses to 2-methoxy-CHol at concentrations up to 100 µM (*Figure 6A and B*), indicating that the 2-methoxy-CHol moiety is indispensable but insufficient for the activation of TRPM8.

These findings raised the question of whether the 2-methoxy-CHol moiety is essential for the binding of the macrolides to TRPM8, or rather affects the ability of the compounds to modulate the gating of TRPM8 following binding. To address this, we performed in silico dockings with everolimus, a close structural analog of rapamycin, in which the hydroxyl group on the cyclohexene ring is substituted by a 2-hydroxyethyl moiety. These revealed that everolimus can bind in the same region of TRPM8 as rapamycin, albeit with reduced binding energy (*Figure 6—figure supplement 1*). In particular, the docking indicated that substituting the hydroxyl group for the 2-hydroxyethyl moiety prevented the formation of a hydrogen bond with the amide of Q861 (*Figure 6—figure supplement 1*). If everolimus can indeed occupy the rapamycin binding site on TRPM8, without strongly activating the channel, the prediction would be that everolimus reduces the effect of rapamycin by competing for the same binding site. This was indeed confirmed in calcium imaging experiments showing that preincubation with 10 µM everolimus, a concentration which by itself does not evoke a substantial calcium signal, significantly inhibited the response to 10 µM rapamycin (*Figure 6C and E*). In contrast, 10 µM everolimus was without any effect on the response to 50 µM menthol (*Figure 6D and E*). Taken together, these findings indicate that macrolide analogs such as everolimus compete with rapamycin for the same binding site but with strongly reduced potency to activate the channel. The lack of effect on channel activation by menthol further confirms that the rapamycin and menthol binding sites are spatially separate and independent.

## Discussion

Our study demonstrates that, in addition to its well-established inhibitory effect on the mTOR signaling pathway (*Abraham and Wiederrecht, 1996*; *Lamming et al., 2013*), rapamycin acts as an agonist of the ion channel TRPM8. Several lines of evidence indicate that the agonistic effect of rapamycin involves a direct interaction with the channel, rather than an indirect effect involving intracellular signaling pathways such as mTOR. First, we found that rapamycin rapidly and reversibly activates TRPM8 currents, not only in the whole-cell mode but also in inside-out patches, indicating a membrane-limited mode of action. Second, we developed STTD NMR methodology, which not only allowed us to demonstrate a direct molecular interaction between rapamycin and TRPM8, but also to identify moieties on rapamycin that interact with the channel. Finally, site-directed mutagenesis steered by molecular docking of rapamycin onto TRPM8 revealed residues that strongly reduce channel activation by rapamycin, while leaving responses to other stimuli intact. Although we cannot exclude that some of these mutations may have an allosteric effect on rapamycin-induced channel gating, the cumulative results are consistent with a model where rapamycin binds in the groove between the voltage sensor-like domain and the pore domain, at a site distinct from the interaction sites of well-studied TRPM8 agonists menthol and icilin.

The effect of rapamycin on TRPM8 occurs at low micromolar concentrations, which contrasts to the low nanomolar concentrations required to inhibit mTOR signaling (*Abraham and Wiederrecht, 1996*; *Varnai et al., 2006*). Plasma concentrations of rapamycin in patients on systemic drug regimens following organ transplantation or for the treatment of lymphangioleiomyomatosis or cancer rarely exceed 100 nM (*Jimeno et al., 2008*; *Trepanier et al., 1998*), rendering it unlikely that substantial rapamycin-induced TRPM8 activation would occur in these subjects. However, rapamycin is also used topically for the treatment of skin diseases, with the use of creams or ointments containing millimolar concentrations of rapamycin. For instance, a 0.2% rapamycin gel (corresponding to approximately 2 mM) has been approved by the FDA for the treatment of facial angiofibromas in patients with tuberous sclerosis complex, and formulations up to 1% have shown efficacy in preclinical and clinical studies for this and other skin disorders, including psoriasis and oral lichen planus (*Aitken et al., 2023*; *Ormerod et al., 2005*; *Soria et al., 2009*). Following such treatments, local rapamycin concentrations

are likely to rise to high-enough levels for activation of TRPM8 in sensory nerve endings of the skin, which could potentially contribute to beneficial effects of the drug in certain skin conditions. Indeed, it is well established that activation of TRPM8 in sensory neurons has anti-inflammatory effects and causes relief of pain and itch (*Liu et al., 2013*; *Palkar et al., 2018*). Further research is required to elucidate the extent of this contribution and its clinical implications.

In addition to rapamycin, we also tested a comprehensive set of related macrolide immunosuppressants for their effect on TRPM8. In line with earlier work, FK506 also robustly activates TRPM8 (*Arcas et al., 2019*), albeit with lower potency than rapamycin, whereas the other tested macrolides had only very modest effects. Overall, these findings indicate that the *trans*-2-methoxycyclohexan-1-ol moiety is crucial for activity at TRPM8. Indeed, substitutions at this site (as for instance in everolimus) almost fully abolish TRPM8-mediated responses, whereas activity is at least partially maintained in compounds where this moiety is retained but with altered macrolactam ring (as in FK506 or ascomycin). These findings indicate that the structural determinants for macrolide activation of TRPM8 differ from the determinants for immunosuppressant activity, which are critically determined by the macrolactam ring (*Chen and Zhou, 2020*). Notably, we found that everolimus can inhibit activation of TRPM8 by rapamycin but not by menthol, indicating that it likely competes with rapamycin for the same binding site.

Overall, our data are consistent with a macrolide binding site on TRPM8, in the groove between voltage sensor-like domain and the pore domain, distinct from the interaction sites of cooling agents and known TRPM8 agonists menthol and icilin, but partially overlapping with the binding site of AITC. The existence of this binding site is also coherent with the finding of Acras et al. showing that TRPM8 sensitivity to FK506/tacrolimus is not affected by Y745H and N799A mutations which eradicate TRPM8 sensitivity to menthol and icilin, respectively. Notably, this groove is also involved in the binding of the antagonist NDNA in the closely related TRPM5 channel. Alignment of TRPM8 with other TRPM channels indicates that some of the key residues for rapamycin activation (D796 and D802) are conserved amongst other TRPM channels, whereas residue Q861 is unique for TRPM8 and G805 is only found in TRPM8 and TRPM2 (*Huffer et al., 2020*). Further studies may elucidate whether the introduction of these residues may transfer rapamycin sensitivity to other TRPM channels. Moreover, structural alignment indicates that the rapamycin binding site in TRPM8 is in a similar location as the vanilloid binding site in TRPV1, although sequence alignment reveals that the key residues involved in binding are not conserved between these more distally related channels (*Cao et al., 2013*; *Huffer et al., 2020*; *Kwon et al., 2021*) The rapamycin binding site may provide a potential clue for the development of drugs that can selectively target TRPM8 for therapeutic purposes, without the side effects associated with other TRPM8 agonists. We acknowledge the limitations of our docking approach, which did not include the lipid bilayer and its interactions with the channel and ligand (*Hughes et al., 2019*). Nevertheless, the integration of docking with experimental validation, such as NMR and detailed calcium imaging and electrophysiology on selected mutants, strengthens the validity of our model. Interestingly, rapamycin also directly activates TRPML1, although information on a possible interaction site on that distally related channel is currently lacking (*Zhang et al., 2019*). We note that none of the key residues involved in activation of TRPM8 by rapamycin are conserved in TRPML1. Moreover, the order of potency of rapamycin analogs differs substantially between TRPML1 and TRPM8. For instance, TRPML1 is potently activated by temsirolimus, whereas this compound is without agonistic effect at TRPM8 (*Zhang et al., 2019*).

Finally, our results also serve as a cautionary note for the use of rapamycin in chemical dimerization studies, a common approach in the study of ion channels (*Suh et al., 2006*; *Varnai et al., 2006*). To exclude a contribution of TRPM8 in such experiments, the use of rapalogs such as everolimus represents a likely safer alternative.

In conclusion, our study reveals a hitherto undiscovered pharmacological property of rapamycin, identifying it as a direct agonist of the TRPM8 ion channel. These findings provide new insights into ligand activation of TRPM8 and may have potential therapeutic implications in the topical use of rapamycin for the treatment of skin diseases.

# Methods

**Key resources table**

| Reagent type (species) or resource | Designation | Source or reference | Identifiers | Additional information |
|---|---|---|---|---|
| Gene (*Homo sapiens*) | TRPM8 | https://www.ncbi.nlm.nih.gov/gene/79054 | 79054 | |
| Strain, strain background (*Mus musculus*, male and female) | C57BL/6 | Janvier Labs | RRID:MGI:2159769 | |
| Strain, strain background (*M. musculus*, male and female) | *Trpm8-/-, -*C57BL/6 | *Dhaka et al., 2007* | | |
| Genetic reagent (*H. sapiens*) | TRPM8 | https://www.ncbi.nlm.nih.gov/nuccore/NM_024080.5/ | NM024080 | |
| Cell line (*H. sapiens*) | HEK293T | ATCC, *Janssens et al., 2016* | ATCC: CRL-3216 | |
| Cell line (*H. sapiens*) | HEK-M8 (HEK293T stably overexpressing hTRPM8) | *Janssens et al., 2016* | | |
| Recombinant DNA reagent | pCAGGSM2-IRES-GFP vector | Addgene | | |
| Recombinant DNA reagent | TransIT-293 transfection reagent | Mirus | Cat#: MIR 2700 | |
| Peptide, recombinant protein | GDNF | Thermo Fisher | Cat#: 450–44 | |
| Peptide, recombinant protein | NT4 | PeproTech | Cat#: 450–04 | |
| Chemical compound, drug | trans-1,2-cyclohexanediol | Sigma-Aldrich | CAS: 1460577 | |
| Chemical compound, drug | Rapamycin (Sirolimus) | LC Laboratories | CAS: 53123889 | |
| Chemical compound, drug | FK506 (Tacrolimus) | LC Laboratories | CAS: 104987113 | |
| Chemical compound, drug | Ascomycin | LC Laboratories | CAS: 104987124 | |
| Chemical compound, drug | Everolimus | LC Laboratories | CAS: 159351696 | |
| Chemical compound, drug | Temsirolimus | LC Laboratories | CAS: 162635043 | |
| Chemical compound, drug | Ridaforolimus | Sigma-Aldrich | CAS: 572924540 | |
| Chemical compound, drug | Zotarolimus | Sigma-Aldrich | CAS: 221877549 | |
| Chemical compound, drug | Pimecrolimus | Sigma-Aldrich | CAS: 137071320 | |
| Chemical compound, drug | Icilin | Sigma-Aldrich | CAS: 36945989 | |
| Chemical compound, drug | Menthol | Sigma-Aldrich | CAS: 2216515 | |
| Software, algorithm | Patchmaster | HEKA Elektronik | RRID:SCR_000034 | |
| Software, algorithm | Autodock 4.2 | https://autodock.scripps.edu/ | RRID:SCR_012746 | |
| Software, algorithm | Gromacs 5.1.4 | https://manual.gromacs.org/documentation/5.1.4/index.html | RRID:SCR_014565 | |
| Software, algorithm | Yasara | https://www.yasara.org | RRID:SCR_017591 | |
| Software, algorithm | PyMOL Molecular Graphics System, Version 2.5.2 | https://www.pymol.org | RRID:SCR_000305 | |
| Software, algorithm | LigPlot | EMBL's European Bioinformatics Institute | RRID:SCR_018249 | |
| Software, algorithm | Topspin, Version 4.0.5 | Bruker Corporation | RRID:SCR_014227 | |
| Software, algorithm | Origin software (9.0 or 2023) | OriginLab | RRID:SCR_002815 | |
| Other | Glass-bottom microwell dish | Fluorodish, World Precision Instruments | Cat#: FD35-100 | Specific dish to measure fluorescence in cell cultures |
| Other | Fura-2-AM | Thermo Fisher Scientific | Cat#: F14185 | Ratiometric fluorescent Ca$^{2+}$ indicator |

*Continued on next page*

*Continued*

| Reagent type (species) or resource | Designation | Source or reference | Identifiers | Additional information |
|---|---|---|---|---|
| Other | Fura-2FF | Santa Cruz | Cat#: sc-218544 | Ratiometric fluorescent $Ca^{2+}$ indicator |
| Other | 96-well black wall/clear-bottom plates | Greiner Bio-One | Cat#: 655090 | Specific plastic ware to measure fluorescence in cell cultures |

## Cell culture and isolation of sensory neurons

Naive HEK293T cells and HEK293T cells stably overexpressing the human TRPM8 (HEK-M8 cells) were cultured at 37 °C in DMEM medium supplemented with 10% fetal bovine serum, 50 U/ml penicillin, 50 µg/ml streptomycin, 10 mM Glutamax, Non-Essential-Amino-Acids (all from Invitrogen/Thermo Fisher). G418 (500 µg/ml) was added to the medium of HEK-M8 cells. HEK293T cells were obtained from ATCC (CRL-3216) and used only up to passage number 25 without further verifying their identity. The cells were tested monthly for the lack of mycoplasma.

Experiments using mice were approved by the KU Leuven Ethical Committee Laboratory Animals under project number P006/2014. All experimental procedures and animal husbandry were conducted following the European Parliament and the Council Directive (2010/63/EU) and national legislation. Sensory neurons from the dorsal root and trigeminal ganglia (DRGs and TGs) were isolated from 8- to 16-week-old wild type (*Trpm8*$^{+/+}$) and TRPM8-deficient (*Trpm8*$^{-/-}$) C57BL/6 mice, as described before (*Vandewauw et al., 2018*). Briefly, mice were euthanized by $CO_2$, DRGs or TGs were isolated and digested with 1 mg/ml collagenase and 2.5 mg/ml dispase dissolved in 'basal medium' (Neurobasal A medium supplemented with 10% FCS) (all from Gibco/Thermo Fisher) at 37 °C for ca. 45–60 min. Digested ganglia were gently washed once in 'basal medium' and twice in 'complete medium' (Neurobasal A medium supplemented with 2% B27 [Invitrogen/Thermo Fisher], 2 ng/ml GDNF [Invitrogen/Thermo Fisher] and 10 ng/ml NT4 [PeproTech, London, UK]) and mechanically dissociated by mixing with syringes fitted with increasing needle gauges. The suspension of sensory neurons was seeded on poly-L-ornithine/laminin-coated glass bottom chambers (Fluorodish, World Precision Instruments, Sarasota, FL USA) and maintained at 37 °C for 24–36 hr before experiments.

## Microfluorimetric intracellular $Ca^{2+}$ imaging

Fluorescent measurement of the cytoplasmic $Ca^{2+}$ concentration in HEK cells and in individual sensory neurons was performed using the fluorescent $Ca^{2+}$-sensitive dye Fura-2 AM in various configurations, as reported earlier (*Kelemen et al., 2021*; *Vandewauw et al., 2018*).

To measure cytoplasmic $Ca^{2+}$ concentration in individual neurons or HEK-M8 cells, a fluorescence microscope-based calcium imaging system was used. Cells were seeded in glass bottom chambers, and the next day, they were loaded with 2 µM Fura-2-AM (Invitrogen/Thermo Fisher Scientific) dissolved in normal $Ca^{2+}$-buffer (150 mM NaCl, 5 mM KCl, 1 mM $MgCl_2 \cdot 6H_2O$, 2 mM $CaCl_2 \cdot 2H_2O$, 10 mM glucose $xH_2O$, 10 mM HEPES, pH 7.4). Fura-2 signals were continuously captured (1 frame/s) using either a CellM (Olympus, Tokyo, Japan) or an Eclipse Ti (Nikon, Tokyo, Japan) fluorescence microscopy system. The cytoplasmic $Ca^{2+}$ concentration was assessed by the ratio of fluorescence measured at $\lambda_{EX1}$: 340 nm, $\lambda_{EX2}$: 380 nm and $\lambda_{EM}$: 520 nm ($F_{340}/F_{380}$). During the measurements, cells were continuously perfused with buffer and different compounds were applied via rapid perfusion. Experiments were performed at room temperature.

To generate dose response curves and investigate rapalogs and other macrolides, we used a fluorescent microplate reader and monitored a population of HEK-M8 cells in each well. HEK-M8 cells were seeded in Poly-L-Lysine HBr coated 96-well black wall/clear-bottom plates (Greiner Bio-One, Frickenhausen, Germany) at a density of 100,000 cells per well in normal cell culture medium and incubated overnight. The next day, cells were loaded with 2 µM Fura-2-AM at 37 °C for 30 min and washed three times with $Ca^{2+}$-buffer. Then, changes in cytoplasmic $Ca^{2+}$ concentration (indicated by the ratio of fluorescence measured at $\lambda_{EX1}$: 340 nm, $\lambda_{EX2}$: 380 nm, $\lambda_{EM}$: 510 nm ($F_{340}/F_{380}$)) were measured using either a FlexStation 3 fluorescent microplate reader (Molecular Devices, Sunnyvale, CA, USA) or an FDSS/µCell analysis system (Hamamatsu Kinetic Plate Imager C13299). In some experiments, fluorescence signals were normalized to baseline values ($F_0$), and results were expressed as

$\Delta F/F_0$ to reflect the relative calcium dynamics. During the measurements, the tested compounds were applied in various concentrations using the built-in pipetting robot of the equipment. In each well, only one given concentration of the agents tested was applied. These measurements were carried out at ambient temperature or at 37 °C.

## Patch clamp electrophysiology

HEK-M8 cells were seeded on poly-L-lysine-coated glass coverslips and transmembrane currents were recorded in the whole-cell or inside-out configurations of the patch-clamp technique using a HEKA EPC-10 amplifier and Patchmaster software (HEKA Elektronik, Lambrecht/Pfalz, Germany). Data were sampled at 5–20 kHz and digitally filtered off-line at 1–5 kHz. Pipettes with final resistances of 2–5 MΩ were fabricated and used to establish a giga-seal access to the membrane. Unless mentioned otherwise, the holding potential was 0 mV and the following voltage step protocol was applied at 0.5 Hz: –80 mV for 200ms,+120 mV for 200ms, and –80 mV for 200ms. In the whole-cell mode, between 70 and 90% of the series resistance was compensated. Whole-cell recordings were performed using an intracellular solution in the patch pipette containing (in mM) 150 NaCl, 5 MgCl2, 5 EGTA and 10 HEPES, pH 7.4 with NaOH. The extracellular solution contained (in mM) 150 NaCl, 1 MgCl2, and 10 HEPES, pH 7.4 with NaOH. In inside-out recordings, the extracellular solution was used as pipette solution, and ligands were applied via the intracellular bath solution.

In patch-clamp experiments where intracellular $Ca^{2+}$ was manipulated through flash photolysis of caged $Ca^{2+}$, the pipette solution contained (in mM): 120 NaCl, 2 Fura-2FF and 20 HEPES, pH 7.4 with NaOH. This solution was further supplemented with 2 mM of the photolysable calcium chelator DM-nitrophen and 1.5 mM $CaCl_2$. Intracellular $Ca^{2+}$ was monitored using a mono-chromator based system consisting of a Polychrome IV monochromator and photodiode detector (TILLPhotonics, Gräfelfing, Germany), controlled by Patchmaster software. Fluorescence was measured during excitation at alternating wavelengths (350 and 380 nm), corrected by subtraction of the background fluorescence before establishing the whole-cell configuration, and represented as $F_{350}/F_{380}$. Rapid photolytic release of $Ca^{2+}$ was achieved by subjecting the cell to brief (~1ms) UV flashes applied from a JML-C2 flash lamp system (Rapp OptoElectronic GmbH, Hamburg, Germany), leading to step-wise, spatially uniform increases in intracellular $Ca^{2+}$ (*Mahieu et al., 2010*).

## Saturation transfer triple-difference (STTD) NMR spectroscopy

[1]H STD NMR experiments were performed on a Bruker Avance Neo 700 MHz spectrometer equipped with a 5 mm z-gradient Prodigy TCI cryoprobe. The temperature was maintained at 298 K. The [1]H reference experiments were run with 256 number of scans, a 1.2 s relaxation delay, and a watergate sequence was utilized for water suppression. These spectra were used to assess sample stability and to scale STD effects (*Figure 2—figure supplement 1*). A train of 50 ms Eburp1 (with $B_1$ field strength of 75 Hz) was employed for selective irradiation of protein [1]H resonances in a total of 3 s. The saturation of aliphatic and aromatic proton regions of TRPM8 was carried out in separate experiments by setting the irradiation (on-resonance) frequencies of –1.8 ppm and 8.8 ppm, respectively, at least 2 ppm away from the resonances of rapamycin to avoid any partial saturation of rapamycin [1]H resonances. Reference spectra were recorded by setting the irradiation frequency off-resonance at –40 ppm. A spin-lock filter of 20ms was applied to reduce the broad signals of the receptor and cell components in the resulting STD spectra. For each STD spectrum (the two on-resonance and one off-resonance), 640 number of scans was acquired in an interleaved fashion to average out any sample and/or spectrometer instability during the measurement, resulting in a total experiment time of 3 hr. All spectra were processed with Topspin version 4.0.5.

NMR samples: $25\times10^6$ HEK293T or HEK-M8 cells/sample were used. Cells were counted by a NovoCyte flow cytometer (Agilent Technologies), and the number of the functional TRPM8 channels was estimated as ca. 5000 channels/cell based on average whole cell currents and single channels conductance available in the literature. 5 µl of rapamycin was added from a 22 mM stock solution resulting in a final concentration of 0.2 mM in $D_2O$ PBS solution, providing ca. $4.8\times10^7$ fold excess of ligand to TRPM8 ion-channels for the STD NMR measurements. The final volume of each sample was 550 µl. NMR data was recorded in three replicates, and a total of 12 individual samples were used to compute three independent STTD spectra (*Figure 2—figure supplement 2*).

## Synthesis of trans-2-Methoxycyclohexan-1-ol

**Scheme 1.** Synthesis of trans-2-Methoxycyclohexan-1-ol.

Trans-2-Methoxycyclohexan-1-ol was synthesized based on the method of *Winstein and Henderson, 1943*. To the solution of *trans*-**1,2-cyclohexanediol** (500 mg, 4.304 mmol) in dry DMF (19 mL) NaH (413 mg, 10.329 mmol, 60 m/m%, 1.2 equiv./OH) was added at 0 °C. After 30 min stirring at that temperature, 308 µl MeI (4.949 mmol, 1.15 equiv.) was added to the mixture and stirred for 24 hr. When the TLC analysis (9:1 $CH_2Cl_2$/MeOH) showed complete consumption of the starting material, the reaction mixture was quenched by the addition of MeOH (2.5 mL). The solution was concentrated under reduced pressure, and the residue was dissolved in $CH_2Cl_2$ (150 mL), washed with $H_2O$, dried on $MgSO_4$ and concentrated. The crude product was purified by column chromatography on silica gel (9:1 $CH_2Cl_2$/MeOH) to give *trans*-**2-methoxycyclohexan-1-ol** (80 mg, 14%) as a colorless syrup. $[\alpha]_D$ −6.54 (*c* 0.26, CHCl$_3$); $R_f$ 0.47 (95:5 $CH_2Cl_2$/MeOH); $^1$H NMR (400 MHz, CDCl$_3$) $\delta$=3.44–3.38 (m, 4 H, OC$H_3$, H-1), 3.10 (s, 1 H, H-1-O$H$), 2.94 (ddd, *J*=10.6 Hz, *J*=8.7 Hz, *J*=4.4 Hz, 1 H, H-2), 2.14–1.03 (m, 8 H, 4 x C$H_2$), ppm; $^{13}$C NMR (100 MHz, CDCl$_3$) $\delta$=85.0 (1C, C-2), 73.6 (1C, C-1), 56.3 (1C, OC$H_3$), 32.2, 28.4, 24.1, 24.0 (4C, 4 x C$H_2$) ppm; MS (UHR ESI-QTOF): *m/z* calcd for $C_7H_{14}NaO_2$, $[M+Na]^+$ 153.0886; found: 153.0885.

## Molecular docking

Initial docking simulations were performed using the cryo-EM structure of the full-length collared flycatcher (*Ficedula albicollis*) TRPM8 (pdb code: 6NR2, 83% sequence identity to human TRPM8) (*Yin et al., 2019*) high-resolution structures of rapamycin (pdb codes: 5FLC, 5GPG) retrieved from the RCSB Protein Data Bank.

Pilot dockings were performed using the Autodock 4.2 software in a 98.0 Å x 98.0 Å x 58.0 Å grid volume, large enough to cover the transmembrane domains of all four subunits of TRPM8 accessible from the extracellular side. The spacing of grid points was set to 1.0 Å and the whole target protein structure and the macrocyclic backbone of the ligand were kept rigid during dockings. 5000 dockings were performed with the above settings and the resultant rapamycin-TRPM8 complexes were clustered and ranked according to the corresponding binding free energies.

Internal flexibility of the rapamycin molecule was assessed through performing simulated annealing using the Gromacs 5.1.4 software package, the CHARMM36 force field, and the GB/SA continuum solvent model. The simulated annealing protocol consisted of 2-ps gradual heating from 50 K to 1050 K, 5-ps equilibration at 1050 K and 5-ps exponential cooling back to 50 K. This protocol was repeated 1000 times for both crystallographic models of rapamycin. The coordinates of the system were stored after each simulated annealing cycle, resulting in a 1000-member ensemble of low-energy structures for both initial rapamycin models. Representative conformations of rapamycin were identified by clustering of the 1000-membered pools, having the macrocycle backbone atoms compared with 1.0 Å RMSD cut-off. Middle structures of the ten most populated clusters, accounting for more than 90% of the total conformational ensemble generated by simulated annealing, were used for further docking studies. To refine initial docking results and to identify plausible binding sites, the above selected rapamycin structures were docked again to the protein target, following the same protocol as above, except for the grid spacing which was set to 0.375 Å in the second pass. The resultant rapamycin-TRPM8 complexes were, again, clustered and ranked according to the corresponding binding free energies. Selected binding poses were subjected to further refinement.

The three most populated and plausible binding poses were further refined by a third pass of docking, where amino acid side chains of TRPM8, identified in the previous pass to be in close contact with rapamycin (<4 Å), were kept flexible. Grid volumes were reduced to these putative binding sites including all flexible amino acid side chains (21.0–26.2 Å x 26.2–31.5 Å x 24.8–29.2 Å). The spacing of grid points was maintained at 0.375 Å. Putative binding modes of rapamycin and TRPM8 were selected from the resultant low energy complexes on the basis of adherence to STTD-NMR data and the number of direct intermolecular contacts.

The initial dockings were further refined upon the publication of Cryo-EM structures of the mouse TRPM8 variant (*Yin et al., 2022*). The hTRPM8 protein sequence Q7Z2W7 was downloaded from the Universal Protein Resource (UniProt) and template Cryo-EM structures of mTRPM8 cold receptor in apo state C1 (PDB ID: 8E4N) and apo state C0 (PDB ID: 8E4P) were downloaded from the Protein Data Bank. The homology modeling was performed with the standard homology modeling protocol implemented in Yasara (https://www.yasara.org). Homology model based on these cryo-EM structures was performed using Swiss model as well with GMQE = 0.71 and sequence identity = 93.74%. All the structures were repaired using the FoldX plugin in YASARA, where RepairPDB identifies those residues which have bad torsion angles, or VanderWaals' clashes, or total energy, and repairs them. Visualization of the molecules was also done with Yasara. Figures were drawn with the open source Pymol (The PyMOL Molecular Graphics System, Version 2.5.2 Schrödinger, LLC; available at https://www.pymol.org). The global docking procedure was accomplished with Autodock VINA algorithms, implemented in Yasara, in which a total of 900 docking runs were set and clustered around the putative binding sites. The grid volumes and parameters were similar to those used for the third pass of docking, described above. The Yasara pH command was set to 7.0, to ensure that molecules preserved their pH dependency of bond orders and protonation patterns. The best binding energy complex in each cluster was stored, analyzed, and used to select the best orientation of the interacting partners. Using a similar docking strategy, we reproduced binding of menthol into the reported menthol binding site based on the cryo-EM structure of TRPM8 in complex with the menthol-analog WS-12 (*Yin et al., 2019*).

## Generation of hTRPM8 mutants

Based on the results of the in silico studies, mutations were introduced at strategic sites within the putative ligand-binding domains. Single amino acids in the full-length human TRPM8 sequence were mutated using the standard PCR overlap extension technique (*Voets et al., 2007*), and the nucleotide sequences of all mutants were verified by DNA sequencing. Constructs were cloned in the bicistronic pCAGGSM2-IRES-GFP vector, transiently expressed in HEK293 cells using TransIT-293 transfection reagent (Mirus, Madison, WI), and characterized using whole-cell patch-clamp electrophysiology and single cell calcium microfluorimetry as described above.

## Chemicals

Rapamycin, FK506, ascomycin, everolimus, and temsirolimus were purchased from LC Laboratories (Woburn, MA USA). Unless indicated otherwise, all other chemicals were obtained from Sigma-Aldrich.

## Data analysis and statistics

Data analysis and statistical tests were performed using Origin software (9.0 or 2023; OriginLab). Data are represented as mean $\pm$ SEM from $n$ cells. One-way ANOVA with a Tukey post-hoc test was used to compare the effect of multiple mutations on TRPM8 ligand responses. Differences in the response profile of somatosensory neurons between genotypes were analyzed using Fisher's exact test. $p < 0.05$ was considered as statistically significant.

Concentration-response curves were fitted using the Hill equation of the form:

$$Response = \frac{Response_{max}}{1 + \left(\frac{EC_{50}}{[C]}\right)^{n_H}}$$

where the calculated parameters are the maximal calcium or current response (Response$_{max}$), the concentration for half-maximal activation (EC$_{50}$), and the Hill coefficient (n$_H$). To quantify ligand-induced

changes in the time course of voltage-dependent current relaxation during voltage steps, mono-exponential functions were fitted to the current traces. While currents in the presence of agonists such as menthol or rapamycin become multiexponential, the monoexponential fits provide a robust quantification of changes in relaxation kinetics.

## Additional information

### Funding

| Funder | Grant reference number | Author |
|---|---|---|
| National Research, Development and Innovation Office | 134725 | Balázs István Tóth |
| National Research, Development and Innovation Office | 134791 | Erika Lisztes |
| National Research, Development and Innovation Office | 137924 | Mihály Herczeg |
| National Research, Development and Innovation Office | GINOP-2.3.2-15-2016-00050 | Balázs István Tóth Tamás Bíró |
| National Research, Development and Innovation Office | GINOP-2.3.3-15-2016-00004 | Katalin E Kövér |
| National Research, Development and Innovation Office | GINOP-2.3.2-15-2016-00044 | Katalin E Kövér |
| Magyar Tudományos Akadémia | János Bolyai Research Scholarship | Balázs István Tóth |
| Innovációs és Technológiai Minisztérium | ÚNKP-22-3-I-DE-324 | Márk Racskó |
| Innovációs és Technológiai Minisztérium | UNKP-22-4-I-DE-87 | Tamás Milán Nagy |
| Innovációs és Technológiai Minisztérium | ÚNKP-21-5-DE-491 | Balázs István Tóth |
| Fonds Wetenschappelijk Onderzoek | G0B9520N | Thomas Voets |
| KU Leuven | C2-TRP | Thomas Voets |
| Queen Elisabeth Medical Foundation for Neurosciences | | Thomas Voets |
| Horizon 2020 Framework Programme | 10.3030/739593 | Balázs István Tóth Thomas Voets |
| H2020 Marie Skłodowska-Curie Actions | 10.3030/955643 | Bahar Bazeli Thomas Voets |
| KU Leuven Research Council | AKUL/19/34 | Thomas Voets |
| Innovációs és Technológiai Minisztérium | ÚNKP-23-3-II-DE-430 | Márk Racskó |

The funders had no role in study design, data collection and interpretation, or the decision to submit the work for publication.

## Author contributions
Balázs István Tóth, Conceptualization, Data curation, Formal analysis, Supervision, Funding acquisition, Validation, Investigation, Visualization, Methodology, Writing – original draft, Project administration, Writing – review and editing; Bahar Bazeli, Tamás Milán Nagy, Data curation, Formal analysis, Validation, Investigation, Visualization, Methodology, Writing – original draft, Writing – review and editing; Annelies Janssens, Data curation, Investigation, Methodology, Project administration; Erika Lisztes, Data curation, Investigation, Methodology; Márk Racskó, Balázs Kelemen, Formal analysis, Investigation, Methodology; Mihály Herczeg, Data curation, Formal analysis, Investigation, Methodology, Writing – original draft; Katalin E Kövér, Conceptualization, Resources, Data curation, Software, Formal analysis, Supervision, Funding acquisition, Validation, Investigation, Visualization, Methodology, Writing – original draft; Argha Mitra, Formal analysis, Investigation, Visualization, Methodology; Attila Borics, Conceptualization, Resources, Data curation, Software, Formal analysis, Supervision, Validation, Investigation, Visualization, Methodology, Writing – original draft, Project administration, Writing – review and editing; Tamás Bíró, Conceptualization, Resources, Supervision, Funding acquisition, Validation, Writing – original draft, Project administration; Thomas Voets, Conceptualization, Resources, Data curation, Software, Formal analysis, Supervision, Funding acquisition, Validation, Visualization, Methodology, Writing – original draft, Project administration, Writing – review and editing

## Author ORCIDs
Balázs István Tóth https://orcid.org/0000-0002-4103-4333
Bahar Bazeli https://orcid.org/0000-0001-9568-2689
Mihály Herczeg https://orcid.org/0000-0002-7938-9789
Attila Borics https://orcid.org/0000-0002-6331-3536
Thomas Voets https://orcid.org/0000-0001-5526-5821

## Ethics
Experiments using mice were approved by the KU Leuven Ethical Committee Laboratory Animals under project number P006/2014. All experimental procedures and animal husbandry were conducted following the European Parliament and the Council Directive (2010/63/EU) and national legislation.

Reviewer #1 (Public review): https://doi.org/10.7554/eLife.97341.3.sa1
Reviewer #2 (Public review): https://doi.org/10.7554/eLife.97341.3.sa2
Reviewer #3 (Public review): https://doi.org/10.7554/eLife.97341.3.sa3
Author response https://doi.org/10.7554/eLife.97341.3.sa4

---

# Additional files

## Supplementary files
MDAR checklist

## Data availability
All data points generated during this study are included in the manuscript and figures. The pdb-file of the rapaymcin binding model is available via Zenodo.

The following dataset was generated:

| Author(s) | Year | Dataset title | Dataset URL | Database and Identifier |
| --- | --- | --- | --- | --- |
| Tóth BI, Bazeli B, Janssens A, Lisztes E, Kelemen B, Herczeg M, Nagy TM, Kövér KE, Mitra A, Borics A, Voets T | 2025 | Direct modulation of TRPM8 ion channels by rapamycin and analog macrolide immunosuppressants | https://doi.org/10.5281/zenodo.17143202 | Zenodo, 10.5281/zenodo.17143202 |

The following previously published datasets were used:

| Author(s) | Year | Dataset title | Dataset URL | Database and Identifier |
|---|---|---|---|---|
| Yin Y, Le SC, Hsu AL, Borgnia MJ, Yang H, Lee S-Y | 2019 | Cryo-EM structure of the TRPM8 ion channel in complex with the menthol analog WS-12 and PI(4,5)P2 | https://doi.org/10.2210/pdb6NR2/pdb | Worldwide Protein Data Bank, 10.2210/pdb6NR2/pdb |
| Lee HB, Lee SY, Rhee HW, Lee CW | 2016 | Co-crystal structure of the FK506 binding domain of human FKBP25, Rapamycin and the FRB domain of human mTOR | https://doi.org/10.2210/pdb5GPG/pdb | Worldwide Protein Data Bank, 10.2210/pdb5GPG/pdb |
| Aylett CHS, Sauer E, Imseng S, Boehringer D, Hall MN, Ban N, Maier T | 2015 | Architecture of human mTOR Complex 1 - 5.9 Angstrom reconstruction | https://doi.org/10.2210/pdb5flc/pdb | Worldwide Protein Data Bank, 10.2210/pdb5FLC/pdb |
| Yin Y, Zhang F, Feng S, Butay KJ, Borgnia MJ, Im W, Lee S-Y | 2022 | The closed C1-state mouse TRPM8 structure in complex with PI(4,5)P2 | https://doi.org/10.2210/pdb8E4N/pdb | Worldwide Protein Data Bank, 10.2210/pdb8E4N/pdb |
| Yin Y, Zhang F, Feng S, Butay KJ, Borgnia MJ, Im W, Lee S-Y | 2022 | Mouse TRPM8 structure determined in the ligand- and PI(4,5)P2-free condition, Class I , C0 state | https://doi.org/10.2210/pdb8E4P/pdb | Worldwide Protein Data Bank, 10.2210/pdb8E4P/pdb |

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
