## [Editor Report · eLife Assessment]

This manuscript presents **valuable** findings showing that rapamycin directly activates the cool-sensing ion channel, TRPM8, acting through a different binding site than other small-molecule cooling agents such as menthol. The use of Ca2+-imaging, electrophysiology, and computational biology provides **solid** evidence to support the finding. The authors also present a novel NMR-based method to help identify details of the binding site interactions. In this revised version, some analysis and the presentation have been corrected and improved. Their findings provide insights into TRP channel pharmacology and may indicate previously unknown physiological effects or therapeutic mechanisms of the immunosuppressant, rapamycin.

---

## [Referee Report · Reviewer #1 (Public review)]

Summary:

In this valuable study, the authors found that the macrolide drug rapamycin, which is an important pharmacological tool in the clinic and the research lab, is less specific than previously thought. They provide solid functional evidence that rapamycin activates TRPM8 and begin to develop an NMR method to measure the specific binding of a ligand to a membrane protein.

Strengths:

The authors use a variety of complementary experimental techniques in several different systems, and their results support the conclusions drawn.

Weaknesses:

The proposed location of the rapamycin binding pocket within the membrane means that molecular docking approaches designed for soluble proteins alone do not provide solid evidence for a rapamycin binding pocket location in TRPM8, but the authors are appropriately careful in stating that the model is consistent with their functional experiments. The novel STTD method is intriguing and supportive of the functional results and docking predictions, but further validation of this method is needed.

Impact:

This work provides still more evidence for the polymodality of TRP channels, reminding both TRP channel researchers and those who use rapamycin in other contexts that the adjective "specific" is only meaningful in the context of what else has been explicitly tested.

Comments on revisions:

The authors have addressed my major concerns from the previous round of revision, and I agree that those things that remain un-done are outside the scope of this manuscript.

---

## [Referee Report · Reviewer #2 (Public review)]

Summary:

Tóth and Bazeli et al. find rapamycin activates heterologously-expressed TRPM8 and dissociated sensory neurons in a TRPM8-dependent way with Ca2+-imaging. With electrophysiology and STTD-NMR, they confirmed the activation is through direct interaction with TRPM8. Using mutants and computational modeling, the authored localized the binding site to the groove between S4 and S5, different than the binding pocket of cooling agents such as menthol. The hydroxyl group on carbon 40 within the cyclohexane ring in rapamycin is indispensable for activation, while other rapalogs with its replacement, such as everolimus, still bind but cannot activate TRPM8. Overall, the findings provide new insights into TRPM8 functions and may indicate previously-unknown physiological effects or therapeutic mechanisms of rapamycin.

Strengths:

The authors spent extensive effort on demonstration that the interaction between TRPM8 and rapamycin is direct. The evidence is solid. In probing the binding site and the structural-function relationship, the authors combined computational simulation and functional experiments. It is very impressive to see that "within" a rapamycin molecule, the portion shared with everolimus is for "binding", while the hydroxyl group in the cyclohexane ring is for activation. Such detailed dissection represents a successful trial in computational biology-facilitated, functional experiment-validated study of TRP channel structural-activity relationship. The research draws the attention of scientists, including those outside the TRP channel field, to previously-neglected effects of rapamycin, and therefore the manuscript deserves broad readership.

Weaknesses:

The significance of the research could be improved by showing or discussing whether a similar binding pocket is present in other TRP channels, and hence rapalogs might bind to or activate these TRP channels. Additionally, while the finding on TRPM8 is novel, it is worthwhile to perform more comprehensive pharmacological characterization, including single-channel recording and a few more mutant studies to offer further insight into the mechanism of rapamycin binding to S4~S5 pocket driving channel opening. It is also necessary to know if rapalogs have independent or synergistic effects on top of other activators, including cooling agents and lower temperature, and its dependence on regulators such as PIP2.

Additional discussion that might be helpful:

The authors did confirm that rapamycin does not activate TRPV1, TRPA1 and TRPM3. But other TRP channels, particularly other structurally-similar TRPM channels, should be discussed or tested. Alignment of the amino acid sequences or structures at the predicted binding pocket might predict some possible outcomes. In particular, rapamycin is known to activate TRPML1 in a PI(3,5)P2-dependent manner, which should be highlighted in comparison among TRP channels (PMID: 35131932, 31112550).

After revision:

I acknowledge that the authors have addressed some of the questions in their revised version. They have explained that additional experiments might be beyond the scope of the current study. I appreciate their effort in doing their best to improve the manuscript and to leave the rest in discussion.

---

## [Referee Report · Reviewer #3 (Public review)]

Summary:

Rapamycin is a macrolide of immunologic therapeutic importance, proposed as a ligand of mTOR. It is also employed as in essays to probe protein-protein interactions.

The authors serendipitously found that the drug rapamycin and some related compounds, potently activate the cationic channel TRPM8, which is the main mediator of cold sensation in mammals. The authors show that rapamycin might bind to a novel binding site that is different from the binding site for menthol, the prototypical activator of TRPM8. These convincing results are important to a wide audience, since rapamycin is a widely used drug and is also employed in essays to probe protein-protein interactions, which could be affected by potential specific interactions of rapamycin with other membrane proteins, as illustrated herein.

Strengths:

The authors employ several experimental approaches to convincingly show that rapamycin activates directly the TRPM8 cation channel and not an accessory protein or the surrounding membrane. In general, the electrophysiological, mutational and fluorescence imaging experiments are adequately carried out and cautiously interpreted, presenting a clear picture of the direct interaction with TRPM8. In particular, the authors convincingly show that the interactions of rapamycin with TRPM8 are distinct from interactions of menthol with the same ion channel.

Weaknesses:

The main weakness of the manuscript was the NMR method employed to show that rapamycin binds to TRPM8. The authors developed and deployed a novel signal processing approach based on subtraction of several independent NMR spectra to show that rapamycin binds to the TRPM8 protein and not to the surrounding membrane or other proteins. In this revised version the authors have strengthened the evidence that the method gives solid results and have improved the clarity of the presentation.

Comments on revisions:

The authors have greatly improved the quality of the presentation of the NMR data and have answered my concerns regarding the new methodology. The manuscript is improved and represents an important contribution.

---

## [Author Response]

The following is the authors’ response to the original reviews.

**Public Reviews:**

**Reviewer #1 (Public Review):**
Summary:In this valuable study, the authors found that the macrolide drug rapamycin, which is an important pharmacological tool in the clinic and the research lab, is less specific than previously thought. They provide solid functional evidence that rapamycin activates TRPM8 and develop an NMR method to measure the specific binding of a ligand to a membrane protein.Strengths:The authors use a variety of complementary experimental techniques in several different systems, and their results support the conclusions drawn.Weaknesses:Controls are not shown in all cases, and a lack of unity across the figures makes the flow of the paper disjointed. The proposed location of the rapamycin binding pocket within the membrane means that molecular docking approaches designed for soluble proteins alone do not provide solid evidence for a rapamycin binding pocket location in TRPM8, but the authors are appropriately careful in stating that the model is consistent with their functional experiments.Impact:This work provides still more evidence for the polymodality of TRP channels, reminding both TRP channel researchers and those who use rapamycin in other contexts that the adjective "specific" is only meaningful in the context of what else has been explicitly tested.
**Reviewer #2 (Public Review):**
Summary:Tóth and Bazeli et al. find rapamycin activates heterologously-expressed TRPM8 and dissociated sensory neurons in a TRPM8-dependent way with Ca2+-imaging. With electrophysiology and STTD-NMR, they confirmed the activation is through direct interaction with TRPM8. Using mutants and computational modeling, the authored localized the binding site to the groove between S4 and S5, different than the binding pocket of cooling agents such as menthol. The hydroxyl group on carbon 40 within the cyclohexane ring in rapamycin is indispensable for activation, while other rapalogs with its replacement, such as everolimus, still bind but cannot activate TRPM8. Overall, the findings provide new insights into TRPM8 functions and may indicate previously unknown physiological effects or therapeutic mechanisms of rapamycin.Strengths:The authors spent extensive effort on demonstrating that the interaction between TRPM8 and rapamycin is direct. The evidence is solid. In probing the binding site and the structural-function relationship, the authors combined computational simulation and functional experiments. It is very impressive to see that "within" a rapamycin molecule, the portion shared with everolimus is for "binding", while the hydroxyl group in the cyclohexane ring is for activation. Such detailed dissection represents a successful trial in the computational biology-facilitated, functional experiment-validated study of TRP channel structuralactivity relationship. The research draws the attention of scientists, including those outside the TRP channel field, to previously neglected effects of rapamycin, and therefore the manuscript deserves broad readership.Weaknesses:The significance of the research could be improved by showing or discussing whether a similar binding pocket is present in other TRP channels, and hence rapalogs might bind to or activate these TRP channels. Additionally, while the finding on TRPM8 is novel, it is worthwhile to perform more comprehensive pharmacological characterization, including single-channel recording and a few more mutant studies to offer further insight into the mechanism of rapamycin binding to S4~S5 pocket driving channel opening. It is also necessary to know if rapalogs have independent or synergistic effects on top of other activators, including cooling agents and lower temperature, and their dependence on regulators such as PIP2.Additional discussion that might be helpful:The authors did confirm that rapamycin does not activate TRPV1, TRPA1 and TRPM3. But other TRP channels, particularly other structurally similar TRPM channels, should be discussed or tested. Alignment of the amino acid sequences or structures at the predicted binding pocket might predict some possible outcomes. In particular, rapamycin is known to activate TRPML1 in a PI(3,5)P2-dependent manner, which should be highlighted in comparison among TRP channels (PMID: 35131932, 31112550).
**Reviewer #3 (Public Review):**
Summary:Rapamycin is a macrolide of immunologic therapeutic importance, proposed as a ligand of mTOR. It is also employed as in essays to probe protein-protein interactions.The authors serendipitously found that the drug rapamycin and some related compounds, potently activate the cationic channel TRPM8, which is the main mediator of cold sensation in mammals. The authors show that rapamycin might bind to a novel binding site that is different from the binding site for menthol, the prototypical activator of TRPM8. These solid results are important to a wide audience since rapamycin is a widely used drug and is also employed in essays to probe protein-protein interactions, which could be affected by potential specific interactions of rapamycin with other membrane proteins, as illustrated herein.Strengths:The authors employ several experimental approaches to convincingly show that rapamycin activates directly the TRPM8 cation channel and not an accessory protein or the surrounding membrane. In general, the electrophysiological, mutational and fluorescence imaging experiments are adequately carried out and cautiously interpreted, presenting a clear picture of the direct interaction with TRPM8. In particular, the authors convincingly show that the interactions of rapamycin with TRPM8 are distinct from interactions of menthol with the same ion channel.Weaknesses:The main weakness of the manuscript is the NMR method employed to show that rapamycin binds to TRPM8. The authors developed and deployed a novel signal processing approach based on subtraction of several independent NMR spectra to show that rapamycin binds to the TRPM8 protein and not to the surrounding membrane or other proteins. While interesting and potentially useful, the method is not well developed (several positive controls are missing) and is not presented in a clear manner, such that the quality of data can be assessed and the reliability and pertinence of the subtraction procedure evaluated.
**Recommendations for the authors:**

**Reviewer #1 (Recommendations For The Authors):**
Major points(1) Given the novelty of the STTD NMR approach, please provide more details and data for the reader.• I would like to see all of the collected spectra so that readers can see and judge the effect sizes for themselves, perhaps as an additional supplementary figure.

We agree with the reviewer that the data transparency of the NMR measurements should be improved. We changed panel C of Figure 2 in the main text and provided all the STD and the computed STDD and STTD spectra recorded on one set of experiments. We carried out additional experimental replicas on new samples and addressed the variability of cell samples by rescaling the STD effects based on reference ^1^H measurements. We provided supplementary spectra of the reference experiments without saturation (Figure S5) and the obtained STTD spectra from the three parallel NMR sessions (Figure S6).

• I appreciate the labels for STDD-1, STDD-2, and STTD on the lower two spectra of Figure 2C. Is the top spectrum from STD-1 or is it prior to saturation? In Figure 2C, what do the x1 and x2 notations on the right-hand side of the spectra indicate?

We showed the top spectrum as an overview and a demonstration of the spectral complexity of the samples. ^1^H experiments were run before the STD measurements to assess the sample quality and stability. The demonstrated spectrum on sample 1 (TRPM8 with rapamycin in HEK cells) was recorded with more transients than the corresponding STDs, thus it is only visually comparable with the difference spectra after scaling (2x). Figure 2 was changed and all the spectra were replaced as mentioned before. All the recorded ^1^H-experiments without saturation including the one removed are now available in the supplementary information (Figure S5).

• The STTD NMR results with WT TRPM8 are consistent with rapamycin binding directly to the channel. Testing whether rapamycin binding observed with STTD NMR is disrupted by one of the most compelling mutations (D796A, D802A, G805A, or Q861A) would be a further test of this direct interaction.

We thank the reviewer for the suggestion and agree that testing the most compelling mutants would be a promising next step. These mutations were generated in plasmid vectors and only transiently transfected into HEK cells. For NMR analysis we would need a high amount of cells stably overexpressing the mutant channels which were not available for experimentation.

• Given that this is not a methods paper, it is probably outside the scope to further validate the STTD NMR measurements by performing parallel ITC, SPR, MST, or radiolabeled ligand experiments. Nevertheless, I would be excited to see such a comparison since STTD NMR appears to have promise as an experimental technique for assessing ligand binding to membrane proteins that does not require large amounts of purified protein or radioactive isotopes.

We agree with the reviewer that additional independent biophysical measurements on the interactions are necessary to further validate the STTD methodology. This paper is a preliminary demonstration of the STTD concept and our group is currently working on the challenges of on-cell NMR (e.g., sample and spectral complexity) and the standardization of the proposed workflow.

(2) Please clarify the methods used to model of rapamycin binding. Docking can be imprecise in TRP channels, even with a sophisticated docking scheme (Hughes et al., 2019, doi: https://doi.org/10.7554/eLife.49572.001).

Thank you for mentioning this point and providing the reference. We have further clarified our methods and included the reference in our discussion, indicating the limitations of our approach.

• As a positive control, does the docking strategy accurately predict binding of known compounds (menthol, icilin, etc.) to TRPM8 consistent with cryo-EM structures?

Yes, the binding site for menthol, based on a similar docking strategy as for rapamycin, is also presented, and matches with predictions from other publications. This is now clarified in the revised manuscript.

• Why was homology modeling to the human sequence used with the mouse structure but not the avian structure?

At this onset of the project, only the avian structure was available, and it was used in the primary docking. Later, to get more precise docking relevant for human TRPM8 pharmacology, we did revert to the then available structure of the mouse ortholog.

• How many rapamycin structural clusters were built, and how many structures were there in each cluster? How many were used? "most populated" is unspecific.

Thank you for your comment. We have added the following highlighted information to the methods section to address your comment:

“Representative conformations of rapamycin were identified by clustering of the 1000-membered pools, having the macrocycle backbone atoms compared with 1.0 Å RMSD cut-off. Middle structures of the ten most populated clusters, accounting for more than 90% of the total conformational ensemble generated by simulated annealing, were used for further docking studies. To refine initial docking results and to identify plausible binding sites, the above selected rapamycin structures were docked again, following the same protocol as above, except for the grid spacing which was set to 0.375 Å in the second pass. The resultant rapamycin-TRPM8 complexes were, again, clustered and ranked according to the corresponding binding free energies. Selected binding poses were subjected to further refinement. The three most populated and plausible binding poses were further refined by a third pass of docking, where amino acid side chains of TRPM8, identified in the previous pass to be in close contact with rapamycin (< 4 Å), were kept flexible. Grid volumes were reduced to these putative binding sites including all flexible amino acid side chains (21.0-26.2 Å x 26.2-31.5 Å x 24.8-29.2 Å).”

However, it is important to clarify that the clusters are not built and their number is not specified by the user. The number of clusters found depends on how similar the structures are in the structural ensemble analyzed by clustering. A high number of clusters indicates a diverse, whereas a low number suggests a uniform structural ensemble. Furthermore, it is arbitrarily controlled by the similarity cutoff specified by the user. If the cutoff is selected well, then the number of structures is different in each cluster. There are some highly populated clusters and a few which only have one structure. The selection of how many cluster representatives are used is usually based on the decision of whether or not the sum of the population of selected clusters sufficiently covers the mapped conformational space.

• Additionally, the rapamycin poses were generated using a continuum solvent model that is unlikely to replicate the conditions existing in the lipid bilayer or in a lipid-exposed binding pocket as is predicted here. It is therefore possible that the rapamycin poses chosen for docking do not represent the physiological rapamycin binding pose, hampering the ability of the docking algorithm to find an appropriate docking pocket.• Furthermore, accurately docking that may bind to membrane-exposed pockets is a challenging problem, particularly because many scoring algorithms, including those employed by Autodock, do not distinguish between solvent-exposed and membrane-exposed faces of the protein. This affects the predicted binding energies.

We appreciate the reviewer's insightful comments. We add a note in discussion part, mentioning these important limitations.

• In Figure 4, it appears that the proposed rapamycin binding pocket is located at the interface between two subunits, but only one is shown. Is there any contact with residues in the neighboring subunit? Based on Figure S4, I assume not, but am unsure.

Based on the estimated distances, we do not think that there are any relevant interactions with residues from neighboring subunits. This is now indicated in the results section.

• Consider uploading the rapamycin-docked model to a public repository such as Zenodo for readers to examine and manipulate themselves

As suggested, the model will be uploaded in a public repository. A link to the file on Zenodo is now included.

(3) Please discuss the spatial location of the proposed rapamycin binding pocket relative to the vanilloid binding pocket in TRPV1.• The mutagenesis indicates that D745, D802, G805, and Q861 are most important for rapamycin sensitivity in TRPM8. Interestingly, the proposed rapamycin binding pocket appears to overlap spatially with the vanilloid binding pocket in TRPV1. Consistent with this, Q861 aligns with E570 in TRPV1, which is a critical residue for resiniferatoxin sensitivity. Indeed, similar to Q861's modeled proximity to the cyclohexyl ring, the hydroxyl group of the vanillyl moity of capsaicin (4DY in 7LR0, for example) is in proximity to E750 in TRPV1. Additionally, searching PubChem by structural similarity suggests that vanillyl head group of the TRP channel modulators capsaicin and eugenol are similar structurally to the trans-2Methoxycyclohexan-1-ol ring. Without overlaying the two structures myself, it is difficult to say more than that, but I encourage the authors to comment on any similarities and differences they observe.• If the proposed rapamycin pocket is indeed similar to the location of the vanilloid binding site, the authors may wish to discuss other TRPM channel structures that show ligands and lipids bound to this pocket because this provides evidence that this pocket influences TRPM channel function. For example, how does the proposed rapamycin binding pocket compare to TRPM8 bound to agonist AITC (PDBID 8e4l), TRPM5 bound to inhibitor NDNA (7mbv), and TRPM2 bound to phosphatidylcholine (6co7)?• Other TRP channel structures with ligands or lipids modeled in this region include TRPV1 bound to resiniferatoxin, capsaicin, or phosphatidylinositol (7l2j, 7l24, 7l2s, 7l2t, 7l2u, 7lp9, 7lpc, 7lqy, 7mz6, 7mz9, 7mza); TRPV3 bound to phosphatidylcholine (7mij, 7mik, 7mim, 7min, 7ugg); TRPV5 bound to econazole (6b5v) or ZINC9155 (6pbf); TRPV6 bound to piperazine (7d2k, 7k4b, 7k4c, 7k4d, 7k4e, 7k4f) or cholesterol hemisuccinate (7s8c); TRPC6 bound to BTDM (7dxf) or phosphatidylcholine (6uza); and TRP1 bound to PIP2 (6pw5).

We thank the reviewer for these valuable insights. We have included some additional discussion highlighting the similarities between the proposed rapamycin binding site and some of the other ligandchannel interactions in the TRP superfamily, in particular the well-known vanilloid binding site in TRPV1. However, to keep the discussion focused, we have not fully discussed all the indicated interactions, to best serve the clarity and scope of the manuscript.

(4) I would like to see negative control calcium imaging and electrophysiology data with untransfected HEK cells to confirm that the observed activation is mediated by TRPM8 to parallel the TRPM8 KO sensory neuron experiments.

This important information is now included in the revised manuscript (Figure S2).

(5) The DM-nitrophen Ca uncaging experiments are an interesting method to test Ca sensitivity of rapamycin, but the results make these experiments more complex to interpret. Ca has been shown to be an obligate cofactor for icilin sensitivity in TRPM8 under conditions where both the internal and external Ca concentrations are tightly controlled (Kuhn et al., 2009, doi: https://doi.org/10.1074/jbc.M806651200), which is necessary because TRPM8 allows Ca permeation through the pore when open. The large icilin-evoked currents in Figure 5A and 5B indicate that the effective intracellular calcium concentration is not zero prior to calcium uncaging, which may be high enough to mask any Ca-dependence of rapamycin that occurs at low Ca concentrations. Given this ambiguity, the inside-out patch clamp configuration would provide more control over the internal and external Ca concentration than is achieved in the Ca uncaging experiments. Because the authors have already demonstrated their ability to perform such experiments (Figure 2 panel B), it would be nice to see tests of Ca dependence using inside-out patch clamp.

As was already shown in Figure 2, Rapamycin activates TRPM8 in inside-out patches, and these experiments were performed using calcium-free cytosolic and extracellular solutions. Note that earlier studies have already shown that icilin activates outward TRPM8 currents in the full absence of calcium: see e.g. Janssens et al. eLife, 2016. Chuang et al. 2004. In the case of Icilin, increased calcium further potentiates the current, which is more prominent for the inward current.

In the Ca uncaging experiments, considering the Kd of DM-nitrophen of 5 nM, we expect that the intracellular calcium concentration before the UV flash would be approximately 15 nM. Taken together, both the inside-out experiments and the flash uncaging experiments confirm that rapamycin responses are not directly regulated by intracellular calcium, contrary to icilin.

(6) Sequence conservation within TRPM channels could be used in combination with the binding pocket model and mutagenesis to predict rapamycin selectivity for TRPM8 over other TRPMs. For example, some important residues, specifically G805 and Q861, are not conserved in TRPM3, which agrees with the lack of rapamycin sensitivity observed in TRPM3 (Figure S1). Further sequence comparison would provide testable hypotheses for future exploration of rapamycin sensitivity in other TRPMs that could validate the proposed binding pocket.

Thank you for the suggestion. We now indicate in the discussion that only some of the key residues are conserved and make suggestions for future studies.

(7) Please unify the color scheme across the figures to improve clarity.• The authors frequently use the colors blue, red, and green to represent menthol and rapamycin in the figures, but they are inconsistent in which one represents menthol and which represents rapamycin. It would be clearer for the audience if, for example, rapamycin is always represented with red and menthol is always represented with blue.

Thank you for pointing this out. We have made the coloring schemes more uniform.

• In Figure 1, panel E, the coloring for Menthol and Pregnenolone Sulfate changes between the TRPM8+/+ and TRPM8-/- panels.

Thank you for pointing this out. We have updated the coloring schemes to ensure consistency between the TRPM8+/+ and TRPM8-/- panels.

• Figure 3 B and E, perhaps color the plot background as a 3-color gradient (blue to white to red) rather than yellow and aqua. Center the white at the WT ratio, keeping the dashed line, with diverging gradients to, for example, blue for mutations that selectively affect menthol sensitivity and red for rapamycin.

Thank you for the suggestion – we have changed the figure accordingly.

• Figure 4 panels A and B use the same color (green) to show two different things (menthol molecule and mutated residues that affect rapamycin sensitivity). It would be clearer for readers to change these colors to agree with a unified color scheme such that, for example, the menthol molecule is colored blue and the rapamycin-neighboring residues are colored red.

Thank you for the suggestion. We have updated the figure to use a unified color scheme, with the menthol molecule now colored green and the rapamycin-neighboring residues colored cyan, to enhance clarity for readers.

• I recommend adding a figure or panel that shows side chains for all mutations, colored by menthol/rapamycin selectivity, as indicated by the functional data in Figure 3B and 3E. This will highlight spatial patterns of the selective residues that are discussed in the text.

Thank you for your suggestion, we added all the side residues in Figure S10.

Minor points(1) It would be nice to have one more concentration data point in the middle of the dose response curve shown in Figure 1 panel B. The response is not saturating at the top or foot of the curve in Figure 1 panel D, precluding a confident fit to a two-state Boltzmann function.

Instead of adding a single data point to this figure, we performed independent measurements on a plate reader system, comparing concentration responses at room temperature and 37 degrees. These data are now included as Figure S1.

(2) The cartoon in Figure 2 panel B should be made more accurate. For example, only the transmembrane helices should be depicted embedded in the membrane, not the whole protein including the intracellular domain. Because the experiment was performed with cells, change the orientation of TRPM8 in the cartoon to show the intracellular domain of the protein facing away from the extracellular side of the membrane where the rapamycin is applied.

Thank you for this comment. We have corrected the cartoon accordingly

(3) Perhaps put the yellow circles under or around the carbon atoms to which the identified hydrogen atoms belong in Figure 2 panel E and Figure 4 panel C. I found it difficult to visualize and compare the STTD NMR results with the predicted binding pocket.

Thank you for the feedback. We have added yellow circles around the carbon atoms corresponding to the identified hydrogen atoms in Figure S9.

(4) Regarding the sentence on p. 12 beginning "In agreement with this notion..."• Include icilin, Cooling Agent-10, and WS-3 as other cooling agents whose sensitivity has been modulated by mutation of Y745• Cryosim-3 responses were not tested in either of the two papers cited; please add citation to Yin et al., 2022, doi: https://doi.org/10.1126/science.add1268.• Other relevant papers include:– Malkia et al., 2009, doi: https://doi.org/10.1186/1744-8069-5-62 which includes molecular docking showing the hydroxyl group of menthol interacting with Y745– Beccari et al., 2017, doi: https://doi.org/10.1038/s41598-017-11194-0 Figure 5 shows disruption of icilin and Cooling Agent-10 sensitivity by Y745A– Palchevskyi et al., 2023, doi: https://doi.org/10.1038/s42003-023-05425-6 Figure 3 shows disruption of icilin, cooling agent-10, WS-3, and menthol sensitivity by Y745A o Plaza-Cayon et al., 2022, https://doi.org/10.1002%2Fmed.21920 Review of TRPM8 mutations• typo: Y754H should be Y745H

Thank you for these suggestions. We have added the above references to the text and corrected the typo.

(5) The authors use the competitive action of everolimus on rapamycin activation as evidence that the different macrolides are binding to the same binding pocket. In addition, prior work showed that Y745H and N799A mutations (which render TRPM8 insensitive to menthol and icilin, respectively) do not affect TRPM8 sensitivity to the structurally-related compound tacrolimus (Arcas et al., 2019). This is consistent with the docking and mutagenesis results presented here.

Thank you for this valuable suggestion. We discuss these data in the revised version.

(6) Rapamycin sensitivity has also been observed in TRPML1 (Zhang et al. 2019, doi: https://doi.org/10.1371/journal.pbio.3000252).

We added a short reference to this interesting finding in the discussion.

(7) The whole-cell currents are very large in several of the electrophysiology experiments (for example Figure 3 panel D and Figure S1), which could lead to artifacts of voltage errors as well as ion accumulation/depletion. However, because this paper is not relying on reversal potential measurements or trying to quantify V1/2, these errors are unlikely to affect the qualitative conclusions drawn.

This is a fair point, but indeed unlikely to affect our main conclusions. Note that we compensated between 70 and 90% of the series resistance, so we don’t expect voltage errors exceeding ~10 mV.

(8) Ligand sensitivity is frequently species-dependent in TRP channels, so it is interesting that multiple species were used here and that both human and mouse isoforms exhibit rapamycin sensitivity. It should be emphasized that human TRPM8 was used in the calcium imaging and electrophysiology experiments, as well as some docking models, while the mouse isoform was used in the sensory neuron experiments and a mutated avian isoform was used for some docking models.

This information is available in the Methods and we believe it is clear for the readers.

(9) Perhaps discuss the unclear mechanism of G805A action in icilin (but not menthol, cold, or praziquantel) sensitivity because it is not in direct contact with the ligand. For example, Yin et al., 2019 propose flexibility allowing Ca binding site and larger binding site for icilin.

Yin et al. (2019) suggests that the G805A mutation impacts icilin sensitivity by influencing the flexibility of the binding site and possibly affecting calcium binding. In our study, we found that G805A significantly reduces rapamycin sensitivity, likely due to its direct role in the rapamycin binding pocket rather than affecting calcium binding. This is now briefly mentioned in the results section.

(10) The Figure S1 legend indicates that n=5 for all panels, so please show normalized population IV curves rather than individual examples. Additionally, it would be interesting to see what happens when each agonist is co-applied with rapamycin. Does rapamycin potentiate or inhibit agonist activation in these channels and/or TRPM8?

We believe that normalized population IVs are not ideal for representing whole-cell currents, considering the substantial variation in current densities. We therefore prefer to show example traces in Figure S3 of the revised version but include mean values of current densities for all tested cells in the text.

While the effects of co-application of rapamycin with activating ligands could be of interest, we consider this somewhat outside the scope of the present manuscript. The combination of HEK293 cell experiments, along with results obtained in WT and TRPM8-deficient mice does, in our opinion, sufficiently describe the selectivity of rapamycin towards TRPM8 compared to other sensory TRP channels.

(11) Figure S1 panel A does not contain units for Rapamycin or AITC concentrations.

Thank you for pointing this out. The units were added to the figure.

(12) It would be nice if the authors characterized the different mutations as predicted to contribute to site 1 (D796, H845, Q861, based on Figure S4), site 2 (D796, M801, F847, and R851), and/or site 3 (F847, V849, and R851).

The indicated mutants were all tested, as shown in Figure 3.

(13) The numbering scheme in Figure S4 does not appear to match the residue numbers in the rest of the paper for certain residues (HIS-844 rather than H845, PHE-846 rather than F847, VAL-848 rather than V849, ARG-850 rather than R851, and GLN-860 rather than Q861), and labels are often overlapping and difficult to see. I also find the transparent spheres very difficult to distinguish from the transparent background, which makes it difficult to appreciate the STTD NMR data overlay.

We apologize for the confusing numbering scheme. The lower numbers refer to the initial docking that was done using the avian TRPM8 ortholog. We have made a newer, clearer version of Figure S4 and inserted as Figure S9.

(14) Please superpose the Ligplots in Figure S5 panels E and F as described in the LigPlus manual (https://www.ebi.ac.uk/thornton-srv/software/LigPlus/manual/manual.html) to facilitate easier comparison.

Thank you for the suggestion. We followed the suggestion to superpose the Ligplots as described but found that the result was visually cluttered and difficult to interpret. To avoid confusion, we instead decided to remove panels E and F from Figure S5, as we believe that the visualization in panels A-D is clear and informative.

(15) Some n values are missing in figure legends.

We checked all legends, and added n numbers were missing.

(16) There is an inconsistent specification of error bars as SEM in the figure legends, though it is specified in methods.A question for my own edification: Here, you have looked at ligand interactions with the protein by saturating the protein resonances and observing transfer to the ligand. Would it be possible to instead saturate lipid or solute resonances and observe transfer to a ligand? I am curious whether this would be one way to measure equilibrium partitioning of ligand into a membrane and/or determine the effective concentration of a ligand in the membrane. Additionally, could one determine whether the compound is fully partitioned into the center of the membrane or just sitting on the surface?

The reviewer highlights an interesting aspect. The widely used WaterLOGSY NMR experiment (doi: 10.1023/a:1013302231549) saturates water molecules then the magnetization is transferred to the ligand of interest. Characteristic changes in ligand resonances are observed in the case of a binding event with proteins. On the other hand, the selective saturation of lipids is -while theoretically possible –technically challenging mainly because of the inherent low signal-dispersion of lipids and peak overlapping with ligand resonances. Additionally, lipid systems are more dynamic compared to proteins and ligand-lipid interactions could be weaker and less specific, significantly affecting the sensitivity of STD experiments.

**Reviewer #2 (Recommendations For The Authors):**
Major:• Is it feasible to test rapamycin on TRPM8 with single-channel recording? This will allow us to better probe the mechanism of rapamycin activation and compare it with menthol, with parameters of singlechannel conductance and maximal open probability.

In our experience, it is very difficult to obtain single-channel recordings from TRPM8. The channel expresses at high densities, typically leading to patches contain multiple channels, making a proper analysis of mean open and closed times very difficult. Therefore, we have decided not to include such measurements in the manuscript.

• The authors classified rapamycin as a type I agonist, the type that stabilizes the open conformation, same as menthol but more prominent. Does that indicate that rapamycin work synergistically (rather than independently) with menthol, because co-application of them can allow them to add to each other in stabilizing the open conformation? I wonder if the authors agree that this could be tested with experiments as in Figure S3, by showing a much more prolonged deactivation with co-application of menthol and rapamycin than applying each alone.

Thank you for the insightful suggestion. We conducted co-application experiments, and our results show that the deactivation time is indeed significantly prolonged when both compounds are applied together compared to each alone. In fact, very little deactivation is seen when both compounds are co-applied, which made it virtually impossible to perform reliable fits to the deactivation time course for the Menthol+Rapamycin condition. Instead, we have now included summary results showing the percentage of deactivation after 100 ms. We included these findings in FigureS8.

• It could be tested whether rapamycin activation of TRPM8 requires or overrides the requirement of PIP2 with inside-out patch by briefly exposing the patch to poly-lysine to sequester PIP2.

This is certainly a good suggestion for further follow-up studies. However, we considered that examination of the (potential) interaction between ligands and PIP2 was outside the scope of the current manuscript.

• Figure 1C suggests that the authors test rapamycin when there is a relatively high baseline TRPM8 activation (prior to rapamycin) activation. This raises the possibility that rapamycin is more a potentiator than an activator. I wonder if the following two experiments could address it: (1) perfuse rapamycin while holding at different membrane potentials, wash-off rapamycin in the solution and quickly (in a few seconds) test the activated current magnitude (before rapamycin dissociation), to compare whether a more depolarized membrane potential (high baseline open probability) allows rapamycin to potentiate more. (2) Perform the experiment at a higher temperature (low baseline open probability) and test whether rapamycin EC50 shifts to the right.

Thank you for the thoughtful suggestion. Overall, we are not really in favor of making a distinction between a potentiator and an activator since it is not really feasible to create a situation where TRPM8 activity is zero. As suggested, we performed the dose response experiment at a higher temperature (37 °C) and observed that rapamycin’s EC_50_ shifts to the right FigureS2. This is similar to what has been observed for menthol on TRPM8 and for many other ligands on other temperature-sensitive TRP channels.

Minor:(1) The author should report hill coefficient together with EC50 when showing dose-responses.

We have added Hill coefficients for all the fits.

(2) In Figure 1 (E, F), it might be clearer to use Venn-diagram to show whether there is overlapping among rapamycin-, menthol-, and cinnamaldehyde-responsive neurons. According to the authors' explanation, we can predict that rapamycin-insensitive, menthol-sensitive neurons should predominantly be cinnamaldehyde-responsive.

Thank you for your suggestion. In these experiments, we applied several agonists and the combination of them would result in a visually crowded Venn diagram difficult to interpret. However, we agree, with the reviewer’s suggestion, and discuss the percentage of the cinnamaldehyde+ neurons in the rapa- menthol+ population in *Trpm8-/-* neurons.

(3) In Figure 3(C), since F847 does not respond to either menthol or rapamycin, it should be excluded from (B). Otherwise it is misleading.

Thank you for pointing this out. To clarify, we have included a calcium imaging trace for the F847 mutant, demonstrating a clear response to rapamycin in FigureS9. This additional data highlights that F847 does respond to rapamycin, albeit with a more modest response amplitude. This is now also clarified in the results section.

(4) The word "potency" in pharmacology usually refers to a smaller EC50 number in dose-dependent experiments. In "Effect of rapamycin analogs on TRPM8" session, the authors use "potency" to refer to response to a single-dose experiment of different compounds. The experiment does not measure potency.

Thank you for pointing out this mistake. We have corrected the text and replaced “potency” with “efficacy”.

(5) "2-methoxyl-" is misspelled in the text body.

We have corrected the typo.

(6) It will be nice to include "vehicle" in Figure 6B, or alternatively normalize all individual traces to vehicle. In Figure 6C and D, everolimus has almost no effect with compared to vehicle, and should not be shown as if it had ~8% in Figure 6B.

We have added the vehicle values to Figure 6B from the same experiments.

**Reviewer #3 (Recommendations For The Authors):**
(1) The NMR method presented here as novel and employed to identify a proposed molecule bound to a membrane protein (TRPM8 in this case) is not well explained and presented. Since several spectra need to be subtracted, the authors should present the raw data and the results of the subtractions step by step. Also, it seems that the height of the peaks in each spectra will be highly variable and thus a reliable criterion employed to scale spectra before subtraction. None of these problems are discussed of described.

The reviewer is right, that the data transparency should be improved and due to the high molecular complexity of the samples the size of the STD effects should be carefully scaled. We carried out additional experimental replicas on new samples and addressed the inherent sample/peak height variability by rescaling the STD effects based on reference ^1^H measurements. We provided supplementary spectra of the reference experiments without saturation (Figure S5) and the computed STTD spectra from three parallel NMR sessions (Figure S6). We changed panel C of Figure 2 in the main text and provided all the STD and the computed STDD and STTD spectra recorded on one set of NMR experiments. We added the following paragraph to the main text: “To address the effect of the inherent variability of cellular samples on peak heights, STD effects were normalized based on the comparison of independent ^1^H experiments (Figure S5). Three STTD replicates were computed, unambiguously confirming direct binding to TRPM8 in two datasets (Figure S6 A,B)”.

Importantly since this signal subtraction method is proposed as a new development, control experiments employing well-established pairs of ligand and membrane protein receptor should be performed to demonstrate the reliability of the method.

We agree with the reviewer, that the STTD experiment as a new development needs further validation, however, this paper is a preliminary demonstration of a new strategy building on the well-established STD and STDD NMR methodologies. Our group is actively engaged in studying additional biological samples to enhance our understanding of the applicability of STTD NMR. These efforts also aim to address challenges such as sample and spectral complexity by refining and standardizing the proposed workflow.

(2) The tail currents shown in supplementary figure 3 are clearly not monoexponential. The fit to a single exponential can be seen to be inadequate and thus the comparison of kinetics of control, rapamycin and menthol is incorrect. At least two exponentials should be fitted and their values compared.

We agree that the decay in the (combined) presence of agonists deviates from a simple monoexponential behavior. While we agree that fitting with two (or more) exponentials would provide a better fit, this also comes with greater variations/uncertainties in the fit parameters. This is particularly the case when inactivation is very slow and incomplete, or when the difference between slow and fast exponential time constants is <5, as seen with rapamycin and rapamycin +menthol. Therefore, we decided to provide monoexponential time constants as a proxy to describe the clear slowing down of activation and deactivation time courses in the presence of Type I agonists.

Also related to this aspect, recordings of TRPM8 currents can not be leak subtracted with a p/n protocol, thus a large fraction of the initial tail current must be the capacitive transient. There is no indication in the methods of how was this dealt with for the fitting of tail currents.

As explained in the methods, capacitive transients and series resistance were maximally compensated. Therefore, we do not agree that a large fraction of the initial tail current must be capacitive. This can also be clearly seen in experiment such as Figure 1C, where the inward tail current is fully abolished in the presence of a TRPM8 antagonist. Likewise, very small and rapidly inactivating tail currents can be seen during voltage steps under control conditions (e.g. Figure S7 and S8 in the revised version).

(3) The docking procedure employed, as the authors show, is not appropriate for membrane proteins since it does not include a lipid membrane. It is not clear in the methods section if the MD minimization described applies only to the rapamycin molecule or to rapamycin bound to TRPM8.It is also not clear if the important residue Q861 (and other residues that are identified as interacting with rapamycin) were identified from dockings or proposed based on other evidence.(4) Identifying amino acid residues that diminish the response to a ligand, does not uniquely imply that they form a binding site or even interact with said ligand. It is entirely possible that they can be involved in the allosteric networks involved in the activating conformational change. This caveat should be clearly posited by the authors when discussing their results.

In our study, we identified several residues that significantly reduce the response to rapamycin when mutated, while retaining robust responses to menthol, which indicates that these mutations do not affect crucial conformational changes leading to channel gating. While our cumulative data suggest that these residues may be involved in direct interaction with rapamycin, we recognize the alternative possibility that they allosterically affect rapamycin-induced channel gating. This is now clearly stated in the first paragraph of the discussion.